# Robust Compressed Sensing MRI with Deep Generative Priors

**Ajil Jalal**[*]
ECE, UT Austin
ajiljalal@utexas.edu

**Marius Arvinte**[*]
ECE, UT Austin
arvinte@utexas.edu

**Giannis Daras**
CS, UT Austin
giannisdaras@utexas.edu

**Eric Price**
CS, UT Austin
ecprice@cs.utexas.edu

**Alexandros G. Dimakis**
ECE, UT Austin
dimakis@austin.utexas.edu

**Jonathan I. Tamir**
ECE, UT Austin
jtamir@utexas.edu

## Abstract

The CSGM framework (Bora-Jalal-Price-Dimakis'17) has shown that deep generative priors can be powerful tools for solving inverse problems. However, to date this framework has been empirically successful only on certain datasets (for example, human faces and MNIST digits), and it is known to perform poorly on out-of-distribution samples. In this paper, we present the first successful application of the CSGM framework on clinical MRI data. We train a generative prior on brain scans from the fastMRI dataset, and show that posterior sampling via Langevin dynamics achieves high quality reconstructions. Furthermore, our experiments and theory show that posterior sampling is robust to changes in the ground-truth distribution and measurement process. Our code and models are available at: https://github.com/utcsilab/csgm-mri-langevin.

## 1 Introduction

Compressed sensing [23, 15] has enabled reductions to the number of measurements needed for successful reconstruction in a variety of imaging inverse problems. In particular, it has led to shorter scan times for magnetic resonance imaging (MRI) [62, 90], and most MRI vendors have released products leveraging this framework to accelerate clinical workflows. Despite their successes, sparsity-based methods are limited by the achievable acceleration rates, as the sparsity assumptions are either hand-crafted or are limited to simple learned sparse codes [72, 73].

More recently, deep learning techniques have been used as powerful data-driven reconstruction methods for inverse problems [49, 68]. There are two broad families of deep learning inversion techniques [68]: end-to-end supervised and distribution-learning approaches. End-to-end supervised techniques use a training set of measured images and deploy convolutional neural networks (CNNs) and other architectures to learn the inverse mapping from measurements to image. Network architectures that include both CNN blocks and the imaging forward model have grown in popularity, as they combine deep learning with the compressed sensing optimization framework, see e.g. [32, 3, 64]. End-to-end methods are trained for specific imaging anatomy and measurement models and show excellent performance in these tasks. However, reconstruction quality is known to suffer when applied out of distribution, and recently has been shown to severely degrade [4, 19] under certain types of natural measurement and anatomy perturbations.

In this paper we study deep learning inversion techniques based on distribution learning. These models are trained without reference to measurements, and so easily adapt to changes in the measurement

---

[*]Ajil Jalal and Marius Arvinte contributed equally to this work.

35th Conference on Neural Information Processing Systems (NeurIPS 2021).

process. The most common family of such techniques, known also as Compressed Sensing with Generative Models (CSGM) [13] uses pre-trained generative models as priors. Generative models are extremely powerful at representing image statistics and CSGM has been successfully applied to numerous inverse problems [13, 34] including non-linear phase retrieval [35], and improved with invertible models [6], sparsity based deviations [21], image adaptivity [42], and posterior sampling [79, 45]. These methods have only recently been applied to MRI and have not yet been shown to be competitive with supervised end-to-end methods. The very recent work [53] trains a StyleGAN for magnitude-only DICOM images but requires the presence of side-information and studies Gaussian, real-valued measurements for reconstruction. The deviation from the true MRI measurement model and the use of magnitude images are known to be problematic when evaluating performance [77]. Another work [54] trained an Invertible Neural Network on complex-valued single-coil MR images and showed very good performance in comparison to sparsity and GAN priors. Untrained and unamortized generators [37] have also been recently explored [19], showing promising results in some cases. Further, [17] studies the harder problem of learning a generative model for a class of images using only partial observations, as first proposed in AmbientGAN [14].

In this paper we train the first score-based generative model [80] for MR images. We show that we can faithfully represent MR images without any assumptions on the measurement system. As a consequence, we are able to reconstruct retrospectively under-sampled MRI data under a variety of realistic sampling schemes. We show that our reconstruction algorithm is competitive with end-to-end supervised training when the test-data are matched to the training data and that it is robust to various out-of-distribution shifts, while in some cases end-to-end methods significantly degrade.

## 1.1  Contributions

- We successfully train a score-based deep generative model for complex-valued, T2-weighted brain MR images without any assumptions on the measurement scheme. When applied to multi-coil MRI reconstruction under the CSGM framework, we achieve competitive performance compared to end-to-end deep learning methods when the test-time data are sampled within distribution.

- We give evidence that posterior sampling should give high-quality reconstructions. First, we show that for any measurements (including the Fourier measurements in MRI) that posterior sampling with the correct prior is within constant factors of the optimal recovery method; second, even if the prior is wrong but gives $\alpha$ mass to the true distribution, we show that posterior sampling for Gaussian measurements is nearly optimal with just an additive $O(\log(1/\alpha))$ loss.

- We empirically show that our approach is robust to test-time distribution shifts including different sampling patterns and imaging anatomy. The former is unsurprising given that our model was trained without knowledge of the measurement scheme. As a consequence, our approach provides a degree of flexibility in choosing scan parameters – a common situation in routine clinical imaging. Perhaps surprisingly, the latter indicates that a specialized training set may offer sufficient regularization for a larger class of images. In contrast, we empirically show that end-to-end methods do not always enjoy the same robustness guarantees, in some cases leading to severe degradation in reconstruction quality when applied out-of-distribution.

- Our method can be used to obtain multiple samples from the posterior by running Langevin dynamics with different random initializations. This allows us to get multiple reconstructions which can be used to obtain confidence intervals for each reconstructed voxel and visualize our reconstruction uncertainty on a voxel-by-voxel resolution. Uncertainty quantification can be incorporated into end-to-end methods, e.g., using variational auto-encoders [24], but this requires changes to the architecture. Our method does not require any modification and multiple reconstruction samplers can be run in parallel.

Our main results are succinctly summarized in Figure 1: we achieve equivalent reconstruction performance using a reduced training set when evaluated in-distribution and are robust when evaluated out-of-distribution.

## 1.2  Related Work

Generative priors have shown great utility to improving compressed sensing and other inverse problems, starting with [13], who generalized the theoretical framework of compressed sensing and restricted eigenvalue conditions [85, 23, 12, 15, 40, 11, 10, 25] for signals lying on the range of

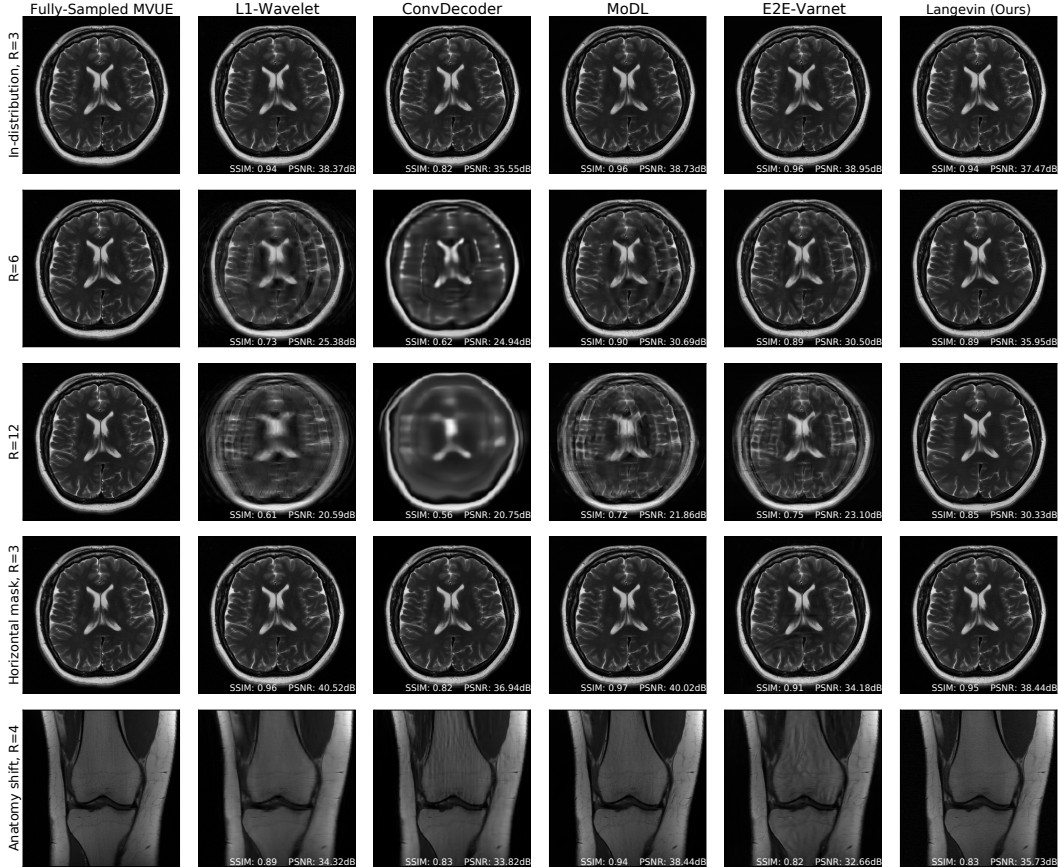

Figure 1: Comparison of reconstruction methods for in-distribution, sampling-shift, and anatomy-shift images. All methods and hyperparameters were optimized on T2-weighted *brain* scans with a vertical sampling mask, and tested at higher accelerations, horizontal masks, and on knee & abdomen scans. Our reconstructions are competitive with state-of-the-art methods, and introduce fewer artifacts out of distribution. All measurements are multicoil k-space from the NYU fastMRI dataset and the supervised baselines are trained from scratch on MVUE targets for a fair comparison.

a deep generative model [29, 55, 81]. Lower bounds in [51, 61, 48] established that the sample complexities in [13] are order optimal. The approach in [13] has been generalized to tackle different inverse problems [47, 35, 7, 71, 60, 63, 74, 9], and different reconstruction algorithms [21, 50, 69, 27, 26, 64, 37, 38, 18]. The complexity of optimization algorithms using generative models have been analyzed in [28, 39, 58, 36]. Our prior work shows that posterior sampling is instance-optimal for compressed sensing [45], and satisfies certain fairness guarantees without explicit information about protected sensitive groups [46].

Using compressed sensing for multi-coil MRI reconstruction has led to a rich body of work in the past two decades [62, 20, 87, 75]. See [22] and the recent special issue [44] for an overview of these methods. Classical approaches impose sparsity in a well-chosen basis, such as the wavelet domain [62], or apply shallow learning that leverages low-level redundancy in the images [72, 73, 93]. Recent research has demonstrated the superior performance of deep neural networks for MR image reconstruction [76, 32, 3, 82, 83]. A broad class of approaches is represented by end-to-end unrolled methods, which use deep networks as learned data priors in the image [3, 32, 82] or k-space domain [84]. Recent work has also investigated the performance of untrained methods [89, 38] for MR reconstruction and has shown competitive results. A much less explored line of research is MR image reconstruction with generative priors. The work in [67] proposes a CSGM-like algorithm that finetunes an entire pre-trained generator that requires a carefully tuned optimization algorithm during inference.

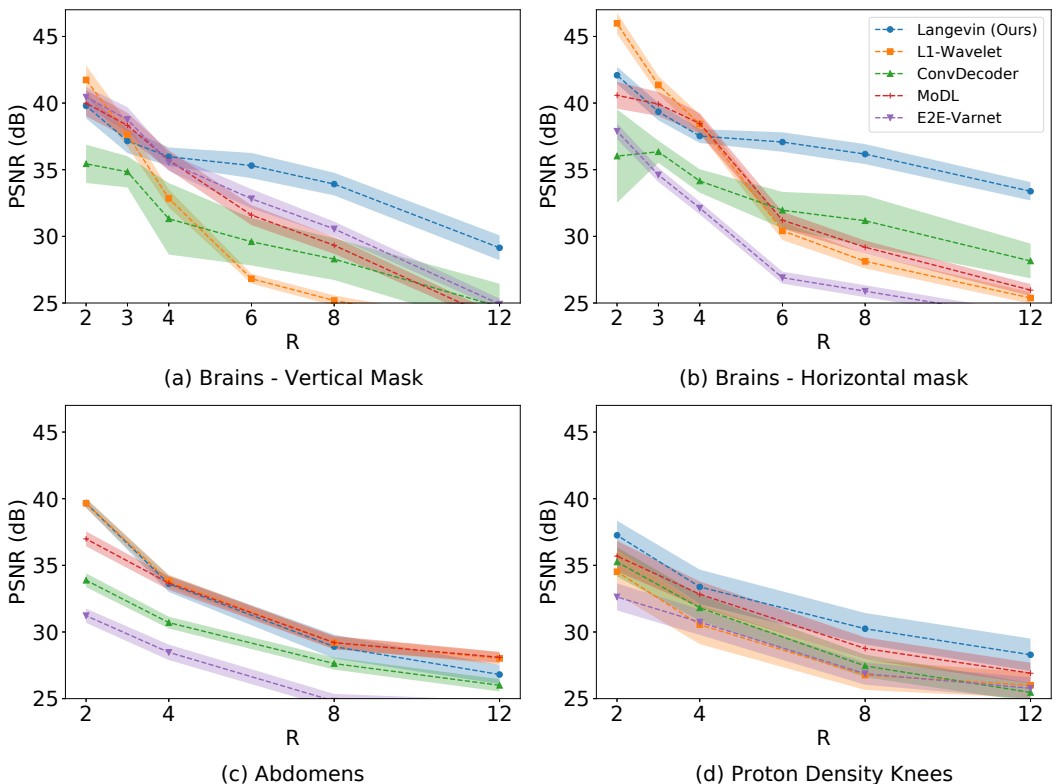

Figure 2: Average test PSNR in various scenarios, across a range of acceleration factors $R$. Higher $R$ indicates a smaller number of acquired measurements. All methods and hyperparameters were optimized on brains with an equispaced vertical mask. Our approach mostly shows the best performance and lowest reconstruction variance both in- and out-of-distribution at test-time. Shaded regions indicate 95% confidence intervals. Note that we trained baselines on MVUE images and hence these numerical values should not be compared with those in literature trained on RSS images (see Appendix A.1 for a more detailed discussion).

## 2 System Model and Algorithm

### 2.1 Multi-coil Magnetic Resonance Imaging

MRI is a medical imaging modality that makes measurements using an array of radio-frequency coils placed around the body. Each coil is spatially sensitive to a local region, and measurements are acquired directly in the spatial frequency, or *k-space*, domain. To decrease scan time, reduce operating costs, and improve patient comfort, a reduced number of k-space measurements are acquired in clinical use and reconstructed by incorporating explicit or implicit knowledge of the spatial sensitivity maps [78, 70, 30]. Formally, the vector of measurements $y_i \in \mathbb{C}^L$ acquired by the i[th] coil can be characterized by the forward model [70]:

$$y_i = PFS_i x^* + w_i, \quad i = 1, ..., N_c, \tag{1}$$

where $x^* \in \mathbb{C}^N$ is the image containing $N$ pixels, $S_i$ is an operator representing the point-wise multiplication of the i[th] coil sensitivity map, $F$ is the spatial Fourier transform operator, $P$ represents the k-space sampling operator, and we assume $w_i \sim \mathcal{N}_c\left(0, \sigma^2 I\right)$ for simplicity. Importantly, note that the same under-sampling operator is applied to all $N_c$ coils.

The acceleration factor $R$ denotes the degree of under-sampling in the $k$-space domain, i.e., $R = N/L$. Due to the multiple coils, the measurements may not be compressive for small $R$. However, due to redundancy between the coils, the measurements are compressive for moderate values of $R$ (even if $N_c \cdot L > N$) [41]. Also note that we use the *true acceleration factor $R$*, and this does not match the values in fastMRI [56] [2] on certain sampling patterns.

---

[2] https://github.com/facebookresearch/fastMRI/blob/main/fastmri/data/subsample.py, line 247 has the fastMRI definition of equispaced acceleration factors.

Given multi-coil measurements $y$, sensitivity maps represented by $S$ and the sampling operator $P$, the goal of MR image reconstruction is to estimate the underlying image variable $x^*$. Prior work formulates this as a regularized optimization problem:

$$\arg\min_x \|y - Ax\|_2^2 + \lambda Q(x), \tag{2}$$

where we use the operator $A \in \mathbb{C}^{M \times N}$ ( with $M = N_c \cdot L$) to subsume the discrete approximation to all linear effects, and $Q$ is a suitably chosen functional prior for the image variable $x$. For example, to enforce a sparsity prior, one can penalize the $\ell_1$ norm in the wavelet representation of $x$ [62]. More recent approaches involve learned regularization terms parameterized by deep neural networks [76, 32, 3]. These models are typically trained *end-to-end* using a fixed training set and certain assumptions about the sampling operator. In the sequel, we present how score-based generative models can be combined with the posterior sampling [45] mechanism to reformulate (2) and achieve good quality reconstructions without any *a priori* assumptions about the sampling scheme.

When k-space is fully sampled at the Nyquist rate and no regularization is applied, the solution to (2) corresponds to the minimum-variance unbiased estimator (MVUE) of $x^*$, denoted by $\hat{x}_{\text{MVUE}}$ [70]. Given fully sampled k-space data, this estimate can act as a reference image for evaluating reconstruction error as well as for end-to-end training. Alternatively, a reference image called the root-sum-of-squares (RSS) estimate can be formed by taking the inverse Fourier transform of each coil and subsequently applying the $\ell_2$ norm for each pixel across the coil dimension, i.e. $\hat{x}_{\text{RSS}} = \sqrt{\sum_{i=1}^{N_c} |(F^H y_i)|^2}$, where $F^H$ is the Hermitian transpose of $F$ (here the inverse DFT). Although the RSS estimate is a biased estimator, it is often used as it does not make any assumptions about the sensitivity maps, which are not explicitly measured by the MRI system. However, even if solving (2) results in perfect recovery of $x^*$, there will be a bias when comparing the result to $\hat{x}_{\text{RSS}}$ and thus the RSS and MVUE cannot be directly compared numerically.

## 2.2 Posterior Sampling

The algorithm we consider is *posterior sampling* [45]. That is, given an observation of the form $y = Ax^* + w$, where $y \in \mathbb{C}^M$, $A \in \mathbb{C}^{M \times N}$, $w \sim \mathcal{N}_c(0, \sigma^2 I)$, and $x^* \sim \mu$, the posterior sampling recovery algorithm outputs $\hat{x}$ according to the posterior distribution $\mu(\cdot|y)$.

In order to sample from the posterior, we use *Langevin Dynamics* [8]. Assuming we have access to $\nabla_x \log \mu(x|y)$, we can sample from $\mu(x|y)$ by running noisy gradient ascent:

$$x_{t+1} \leftarrow x_t + \eta_t \nabla_{x_t} \log \mu(x_t|y) + \sqrt{2\eta_t}\,\zeta_t, \quad \zeta_t \sim \mathcal{N}(0,1). \tag{3}$$

Prior work [8] has shown that as $t \to \infty$ and $\eta_t \to 0$, Langevin dynamics will correctly sample from $\mu(x|y)$. In practice, vanilla Langevin Dynamics are slow to converge. Hence, the work in [79] proposes *annealed* Langevin Dynamics, where the marginal distribution of $x$ at iteration $t$ is modelled as $\mu_t = \mu * \mathcal{N}(0, \beta_t^2)$ and the generative model is trained to estimate the score function $f(x_t; \beta_t) := \nabla_{x_t} \log((\mu * \mathcal{N}(0, \beta_t^2))(x_t))$.

Since the distribution of $y|x^*$ is Gaussian in Eqn (2), we obtain $\nabla_{x_t} \log \mu(y|x_t) = \frac{A^H(y - Ax_t)}{\sigma^2}$. We find that it is also helpful to anneal this term, and we set it to $\frac{A^H(y - Ax_t)}{\sigma^2 + \gamma_t^2}$, where $\gamma_t \to 0$ is a decreasing sequence. An application of Bayes' rule gives: $\nabla_{x_t} \log \mu(x_t|y) = f(x_t; \beta_t) + \frac{A^H(y - Ax_t)}{\sigma^2 + \gamma_t^2}$.

Putting everything together, our final algorithm is: for $x_0 \sim \mathcal{N}_c(0, I)$ and for all $t = 0, \cdots, T - 1$,

$$x_{t+1} \leftarrow x_t + \eta_t \left( f(x_t; \beta_t) + \frac{A^H(y - Ax_t)}{\gamma_t^2 + \sigma^2} \right) + \sqrt{2\eta_t}\,\zeta_t, \quad \zeta_t \sim \mathcal{N}(0; I). \tag{4}$$

Note that the parameters $T, \{\beta_t\}_{t=0}^{T-1}$ were fixed during training of the generative model, and hence the only hyperparameters during inference are $\{\eta_t\}_{t=0}^{T-1}, \sigma$ and $\{\gamma_t\}_{t=0}^{T-1}$. Scripts in our codebase describe hyperparameter values used in our experiments.

# 3 Theoretical Results

**Background and Notation.** We first introduce background and notation required for our theoretical results. $\|\cdot\|$ refers to the $\ell_2$ norm. In this section alone, for simplicity of exposition, we will assume that all matrices and vectors are real valued.

For two probability distributions $\mu, \nu$ on some normed space $\Omega$, and for any $q \geq 1$, the Wasserstein-$q$ [91, 5] and Wasserstein-$\infty$ [16] distances are defined as:

$$\mathcal{W}_q(\mu,\nu) := \inf_{\gamma \in \Pi(\mu,\nu)} \left( \mathbb{E}_{(u,v)\sim\gamma} [\|u-v\|^q] \right)^{1/q}, \quad \mathcal{W}_\infty(\mu,\nu) := \inf_{\gamma \in \Pi(\mu,\nu)} \left( \gamma\text{-}\operatorname*{ess\,sup}_{(u,v)\in\Omega^2} \|u-v\| \right).$$

where $\Pi(\mu,\nu)$ denotes the set of joint distributions whose marginals are $\mu, \nu$. The above definition says that if $\mathcal{W}_\infty(\mu,\nu) \leq \varepsilon$, and $(u,v) \sim \gamma$, then $\|u-v\| \leq \varepsilon$ almost surely.

The $(\varepsilon,\delta)-$*approximate covering number* [45], is defined as the smallest number of $\varepsilon$-radius balls required to cover $1 - \delta$ mass under a distribution.

**Definition 3.1** (($\varepsilon,\delta$)-approximate covering number)**.** *Let $\mu$ be a distribution on $\mathbb{R}^N$. For some parameters $\varepsilon > 0, \delta \in [0,1]$, the $(\varepsilon,\delta)$-approximate covering number of $\mu$ is defined as*

$$\mathrm{Cov}_{\varepsilon,\delta}(\mu) := \min \left\{ k : \mu \left[ \cup_{i=1}^k B(x_i,\varepsilon) \right] \geq 1 - \delta, x_i \in \mathbb{R}^N \right\},$$

*where $B(x,\varepsilon)$ is the $\ell_2$ ball of radius $\varepsilon$ centered at $x$.*

**Distributional robustness under Gaussian measurements.** First, we consider mismatch between the ground-truth distribution, denoted by $\mu$, and the generator distribution, denoted by $\nu$. Prior work [45] has shown that if (i) $\mathcal{W}_q(\mu,\nu) \leq \varepsilon$ for some $q \geq 1$ and (ii) we are given $M \geq O(\log \mathrm{Cov}_{\varepsilon,\delta}(\mu))$ Gaussian measurements, then posterior sampling with respect to $\nu$ will recover $x^* \sim \mu$ up to an error of $\varepsilon/\delta^{1/q}$ with probability $1 - \delta$. Closeness in Wasserstein distance is a reasonable assumption in certain examples, such as when $\mu$ is the distribution of celebrity faces and $\nu$ is the distribution of a generator trained on FlickrFaces [52]. However, this assumption is unsatisfactory when we consider distributions of abdominal and brain MR scans, for example, since images of these anatomies look entirely different.

We define the following weaker notion of divergence between distributions. Informally, this new definition tells us that $\nu$ and $\mu$ are "close" if they can each be split into components which are close in $\mathcal{W}_\infty$ distance, such that the close components contain a sufficiently large fraction under $\nu$ and $\mu$. Formally, this is defined as:

**Definition 3.2** (($\delta,\alpha$)-$\mathcal{W}_\infty$ divergence)**.** *For two probability distributions $\nu$ and $\mu$, and parameters $\delta, \alpha \in [0,1]$, the $(\delta,\alpha)$-$\mathcal{W}_\infty$ divergence is defined as*

$(\delta,\alpha)$-$\mathcal{W}_\infty(\mu,\nu) := \inf\{\varepsilon \geq 0 :$

$\exists \mu', \mu'', \nu', \nu'' \in \mathcal{M}(\mathbb{R}^N) \text{ s.t. } \mu = (1-\delta)\mu' + \delta\mu'', \nu = (1-\alpha)\nu' + \alpha\nu'', \mathcal{W}_\infty(\mu',\nu') = \varepsilon.\}$

Lemma B.1 highlights that this is a strict generalization of Wasserstein distances, in the sense that closeness in Wasserstein distance implies closeness in this new divergence.

Since the $(\delta,\alpha)$-$\mathcal{W}_\infty$ divergence is a generalization of Wasserstein distances, it is not clear that the main Theorem in [45] holds for distributions that are close in this new divergence. The following result shows a rather surprising fact: if $(\delta,\alpha)$-$\mathcal{W}_\infty(\mu,\nu) \leq \varepsilon$ then posterior sampling with $M = O\left(\log\left(\frac{1}{1-\alpha}\right) + \log \mathrm{Cov}_{\varepsilon,\delta}(\mu)\right)$ measurements will still succeed with probability $\geq 1 - O(\delta)$.

**Theorem 3.3.** *Let $\delta, \alpha \in [0,1]$, and $\varepsilon > 0$ be parameters. Let $\mu, \nu$ be arbitrary distributions over $\mathbb{R}^N$ satisfying $(\delta,\alpha)$-$\mathcal{W}_\infty(\mu,\nu) \leq \varepsilon$. Let $x^* \sim \mu$ and suppose $y = Ax^* + w$, where $A \in \mathbb{R}^{M \times N}$ and $w \in \mathbb{R}^M$ are i.i.d. Gaussian normalized such that $A_{ij} \sim \mathcal{N}(0, 1/M)$ and $w_i \sim \mathcal{N}(0, \sigma^2/M)$, with $\sigma \gtrsim \varepsilon$. Given $y$ and the fixed matrix $A$, let $\widehat{x}$ be the output of posterior sampling with respect to $\nu$.*

*Then for $M \geq O\left(\log\left(\frac{1}{1-\alpha}\right) + \min(\log \mathrm{Cov}_{\sigma,\delta}(\mu), \log \mathrm{Cov}_{\sigma,\delta}(\nu))\right)$, there exists a universal constant $c > 0$ such that with probability at least $1 - e^{-\Omega(M)}$ over $A, w$,*

$$\Pr_{x^*\sim\mu, \widehat{x}\sim\nu(\cdot|y)} [\|x^* - \widehat{x}\| \geq c(\varepsilon + \sigma)] \leq \delta + e^{-\Omega(M)}.$$

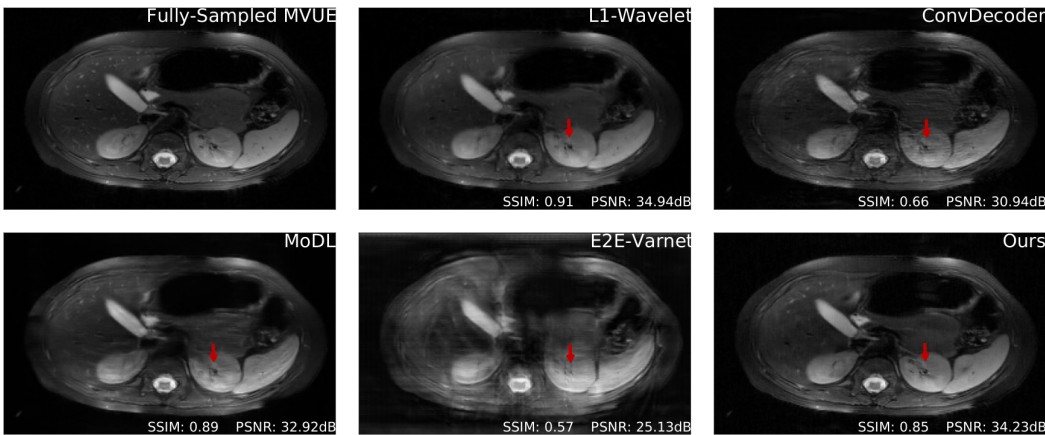

Figure 3: Comparative reconstructions of a 2D abdominal scan with uniform random under-sampling in the horizontal direction at $R = 4$. None of the methods were trained to reconstruct abdomen MRI. Our method uses a score-based generative model trained on brain images (as explained) and obtains good reconstructions. The red arrows indicate missing details or artifacts in the kidney structure.

For our running example of $\nu$ being a generator trained on brain scans, and $\mu$ the distribution of abdominal scans, we can set $\nu'$ to be the distribution of our generator restricted to abdominal scans, and we can let $\mu'$ be the distribution restricted to "inliers" in $\mu$. This shows that even if our generator places an *exponentially small* probability mass(i.e., $1 - \alpha \ll 1$) on the set of abdominal scans, we can still recover abdominal scans with a *polynomial additive* increase in the number of measurements (i.e., $\log(1/(1 - \alpha))$).

**Near-optimality under arbitrary measurement processes.** The previous result required Gaussian matrices to handle the distribution shift. Our next result shows that for an *arbitrary* measurement process, and assuming that there is no distribution shift between the generator and the ground truth distribution, posterior sampling is almost the best algorithm for this *fixed* measurement process. This result also shows that posterior sampling is good with respect to *any* metric.

**Theorem 3.4.** *Let $d(\cdot, \cdot)$ be an arbitrary metric over $\mathbb{R}^N \times \mathbb{R}^N$. Let $x^* \sim \mu$ and let $y = \mathcal{A}(x^*)$ be measurements generated from $x^*$ for some arbitrary forward operator $\mathcal{A} : \mathbb{R}^N \to \mathbb{R}^M$. Then if there exists an algorithm that uses $y$ as inputs and outputs $x'$ such that*

$$d(x^*, x') \leq \varepsilon \text{ with probability } 1 - \delta,$$

*then posterior sampling $\widehat{x} \sim \mu(\cdot | y)$ will satisfy*

$$d(x^*, \widehat{x}) \leq 2\varepsilon \text{ with probability } \geq 1 - 2\delta.$$

**Remark on combining these results.** Our theoretical results above show that posterior sampling is (1) highly robust to distribution shift under Gaussian measurements, and (2) accurate with arbitrary measurements without distribution shift. A natural hope would be to combine these two results and show that it is robust to distribution shift under Fourier measurements. Unfortunately, this is *not* true for general distributions: for example, if $\mu$ and $\nu$ are both random distributions over Fourier-sparse signals, then Fourier measurements will usually give zero information about the signal, so cannot convince the sampler to sample near $\mu$ rather than $\nu$.

## 4    Experimental Results

We perform retrospective under-sampling in all experiments, i.e., given fully-sampled k-space measurements from the NYU fastMRI [56, 94] and Stanford MRI [1] datasets, we apply sampling masks and evaluate the performance of all considered algorithms on the reconstructed data. Depending on scan parameters (e.g., 3D scans for the Stanford knee data in Appendix F), we appropriately slice and sample the data in the proper dimension so as to not commit any inverse crime [31, 77].

We first highlight that an advantage of the proposed approach is the invariance to the sampling scheme during training. In contrast, this is a design choice that must be made for supervised end-to-end methods, which here were trained on equispaced, vertical sampling masks, following the fastMRI 2020 challenge guidelines [94, 66]. As our results show, this affords us a significant degree of robustness across a wide distribution of sampling masks during inference.

We train a score-based model, NCSNv2 [80], on a small subset of scans from the NYU fastMRI brain dataset. Specifically, we train using T2-weighted images at a field strength of 3 Tesla for a total of 14,539 2D training slices. We calculate the MVUE from the fully sampled data and use the ESPIRiT algorithm [87, 43] applied to the fully-sampled central portion of k-space to estimate the sensitivity maps. The backbone network for our model is a RefineNet [59]. Since the generator's output is expected to be complex-valued, we treat the real and imaginary parts as separate image channels. Details about the architectures are given in Appendix G.

We use an $\ell_1$-Wavelet regularized reconstruction algorithm [62] as a parallel imaging and compressed sensing baseline. This aims to solve the optimization problem given in (2) with $Q(x) = ||Wx||_1$, where $W$ is a 2D Wavelet transform. We use the publicly available implementation from the BART toolbox [88, 86] and optimize the regularization hyper-parameter using the same subset of samples from the brain dataset that was used to train our method. We find that $\lambda = 0.01$ performs the best on the training data and use this value for all experiments. We consider three different deep learning baselines: MoDL [3], E2E-VarNet [82], and the ConvDecoder architecture [19].

We train the MoDL and E2E-VarNet baselines *from scratch* on the same training dataset as our method, at acceleration factors $R = \{3, 6\}$ and equispaced under-sampling, with a supervised SSIM loss on the magnitude MVUE image, for 40 and 15 epochs, respectively, using a batch size of 1. For the ConvDecoder baseline, we use the architecture for brain data in [19] that outputs a complex image estimate and optimize the number of fitting iterations on a subset of samples from the training data. We find that 10000 iterations are sufficient to reach a stable average performance at $R = 3$. Put together, all of our baselines are tailored to estimate the complex image $x$, thus all comparisons are fair. We evaluate reconstruction performance using the complex MVUE of the fully sampled data as a reference image and measure the peak signal-to-noise ratio (PSNR) and structural similarity index (SSIM) [92] between the absolute values of the reconstruction and ground-truth MVUE images.

## 4.1 In-Distribution Performance

In this experiment, we test all models using the same forward model that matches the training conditions for the baselines: vertical, equispaced sampling patterns. Examples of various sampling patterns are shown in Appendix C.

Figure 1 (top three rows) shows qualitative results and Figures 2a & 5a respectively show PSNR & SSIM values, for the case where there is no mismatch between the training and inference sampling patterns. As the baselines were trained to maximize SSIM at $R = 3$ & 6, we see that they achieve better SSIM scores than us at these accelerations, although there is clear aliasing in the baselines at $R = 6$. We achieve better PSNR values at these accelerations, which supports the claim that our method does not overfit to a particular metric (Theorem 3.4). This also highlights the importance of qualitative evaluations in medical image reconstruction and the limitations of existing image quality metrics [65]. From the third row of Figure 1, and Figures 2a & 5a, we notice that our method surpasses baselines at higher accelerations.

We find that $\ell_1$-Wavelet suffers both qualitatively and quantitatively at high acceleration factors, while the ConvDecoder is also a competitive architecture, but incurs a large computational cost. When benchmarked on an NVIDIA RTX 2080Ti GPU, our method takes 16 minutes and 0.95 GB of memory to reconstruct a high-resolution brain scan, whereas the ConvDecoder takes longer than 80 minutes and 6.6 GB of memory. While our method is limited by the inference time and is not in the range of end-to-end models (where reconstruction takes at most on the order of seconds and 3.5 GB of memory), multiple scans can be reconstructed in parallel due to the reduced memory footprint.

## 4.2 Out-of-Distribution Performance

**Test-time sampling pattern shifts.** Here we consider shifts in the forward sampling operator at test-time, while still evaluating on the same anatomy as the training conditions. We measure

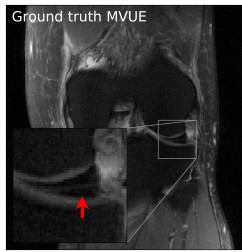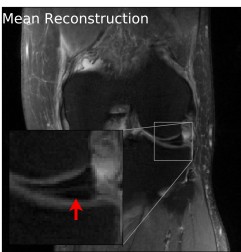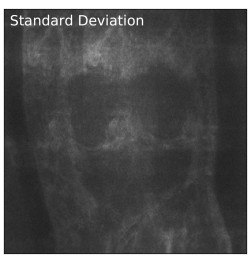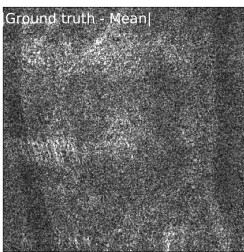

Figure 4: Our method successfully recovers fine details and can provide an estimate of the reconstruction error. The left column shows a knee from the fastMRI dataset, along with an annotated meniscus tear (indicated by red arrow in zoomed inset). Given measurements at an acceleration factor of $R = 4$, we obtain $48$ independent reconstructions via posterior sampling. The second column shows the pixel-wise average of reconstructions, the third column shows the pixel-wise standard deviation, and the fourth column shows the magnitude of the error between the ground truth and the mean reconstruction. Note that our generative prior has never seen such pathology, as it was trained on T2-weighted brain scans.

robustness by evaluating the average incurred performance loss when the sampling pattern changes. Recall that our proposed approach does not use any explicit information about the sampling pattern $P$ during training, hence we anticipate the highest degree of robustness.

Figure 1 (fourth row) shows qualitative reconstructions when the measurements are obtained from an equispaced, horizontal sampling mask, with an acceleration factor $R = 3$. It can be observed that the reconstructions output by E2E-VarNet show aliasing artifacts. Based on the statistical results in Figure 2b & 5b, our method retains its performance.

Furthermore, this experiment reveals that MoDL is more robust to this type of mask shift when compared to E2E-VarNet, even though it uses a smaller network. This is explained by the fact that E2E-VarNet does not use external sensitivity map estimates, but uses a deep neural network for end-to-end map estimation. While this improves performance on in-distribution samples, the performance drop is strong evidence that accurate sensitivity map estimation is vital for robust generalization, and both our proposed approach and MoDL benefit from the external ESPIRiT algorithm, which is compatible with different sampling patterns.

We do note that retrospectively flipping the horizontal and vertical sampling direction is not necessarily representative of prospective sampling in the horizontal direction due to the discrete nature of the phase encoding direction in MRI, and this may contribute to the higher scores compared to the vertical mask experiments.

**Test-time anatomy shifts.** We now consider the more difficult problem of reconstructing different anatomies than the ones seen during. This was previously investigated in [19], which concluded that all methods suffer a drastic shift due to the various changes in scan parameters between body parts. In contrast to prior work, our main finding is that the proposed score-based model retains a significant degree of robustness under these shifts, and outputs excellent qualitative reconstructions. In some cases, some end-to-end methods retain robustness as well.

Figures 2c & 5c show PSNR and SSIM scores obtained on reconstructed abdominal scans obtained from [1] at different acceleration factors. This represents both an anatomy and sampling pattern shift, and it can be seen that our method, MoDL, and the $\ell_1$-Wavelet algorithm retain their competitive advantage, while the ConvDecoder and E2E-VarNet suffer severe performance losses. Figure 3 further shows a qualitative comparison of a reconstructed abdominal scan at $R = 4$, with highlighted artifacts. Appendix E shows another abdomen scan.

Finally, Figures 2d & 5d show PSNR and SSIM scores obtained on fastMRI knee reconstructions, while Figure 1 (bottom row) shows the accompanying qualitative plots. This anatomy is challenging especially because of the poor signal-to-noise ratio conditions, which can be seen even in the ground-truth image. It can be noticed that this is the most severe shift for all methods, but our approach still shows the best performance at $R = 2, 4$ and a significantly lower variance. Appendix D shows more examples of knee reconstructions with and without fat suppression, and Figure 20 shows metrics on fat suppressed knees.

### 4.3 Uncertainty Estimation

Our method can also provide uncertainty estimates for each reconstructed pixel by running multiple reconstruction samplers. For a given observation $y$, we can obtain independent samples $\widehat{x}_1, \cdots, \widehat{x}_K \sim \mu(\cdot|y)$, for $K$ sufficiently large. Now, using the conditional mean estimate $\bar{x} = \sum_{i=1}^{K} \widehat{x}_i/K$, we can compute the pixel-wise standard deviation $\sqrt{\sum_{i=1}^{K} |\widehat{x}_i - \bar{x}|^2/K}$, and this gives an estimate of the error in each pixel. As shown in Fig 4, the pixel-wise standard deviation is a good estimate of the ground truth error $|x^* - \bar{x}|$. Additionally, notice that the reconstructions are able to recover fine details such as the annotated meniscus tear[3] in Fig 4 and predict low uncertainty for these features.

Figure 17 in Appendix D shows another example of an annotated meniscus tear. Figures 18 and 19 show comparisons with baselines on the same examples.

### 4.4 Radiologist Study

We have conducted a preliminary blind assessment of overall image quality with two board-certified radiologists and one faculty member who uses neuroimaging for their research. These experts were *not* involved in our research. We have found that our algorithm was ranked best for knee scans, and tied with the baselines for abdominal and brain scans, supporting our robustness claims in the paper. For more details, please see Appendix H.

## 5 Limitations

We reported PSNR and SSIM values as they are correlated with radiologist evaluation upto an extent, and our preliminary radiologist study in Section 4.4 suggests the feasibility of clinical adoption. These metrics do not capture the needs of real-world radiologists, and a more detailed study is required before the proposed techniques can be clinically adopted.

Though promising, our initial results were still limited to fast spin-echo imaging only and all data were retrospectively under-sampled. Further study is required to demonstrate prospective performance in a larger body of heterogeneous MRI data. Our method also currently requires a high compute cost at inference time, as well as the need for a pre-trained generative model. Clinical use requires fast reconstruction in addition to fast scanning. Future work should investigate whether score-based models can be trained without a fully-sampled training set as well as investigate approaches to reducing computation time.

Finally, there are potential issues related to discrimination. Specifically, it is possible that the quality of the reconstructed images varies across protected attributes, such as gender or race [57].

## 6 Conclusions

This paper reports the first successful application of the CSGM framework for robust multi-coil MR image reconstruction under realistic sampling conditions, and provides theoretical evidence for the robustness of posterior sampling. Our score-based model was trained on a small subset of brain MRI scans without any explicit information about the sampling scheme. This shows state-of-the-art performance under severe distributional shifts, making our model applicable in a wide range of clinical settings.

Our method shows a considerable degree of generalization to out-of-distribution samples such as abdomen and knee MRI, even when trained exclusively on brain MRI. Notably, these scans were acquired using different MRI vendors with different pulse sequence parameters and at different institutions. We postulate that adding a small set of diverse training samples to our generative model could further improve robustness, and we hypothesize that these samples may not necessarily be restricted to MR images.

The results presented in this work represent an important step to applying deep learning models in the clinic, as there is a natural variation in sampling, image orientation, receive coils, scanner hardware, and anatomy in clinical practice.

---

[3]`https://discuss.fastmri.org/t/219`

# 7 Acknowledgements

Ajil Jalal, Giannis Daras and Alex Dimakis have been supported by NSF Grants CCF 1763702, 1934932, AF 1901281, 2008710, 2019844, the NSF IFML 2019844 award as well as research gifts by Western Digital, Interdigital, WNCG and MLL, computing resources from TACC and the Archie Straiton Fellowship.

Eric Price has been supported by NSF Award CCF-1751040 (CAREER), NSF Award CCF-2008868, and NSF IFML 2019844.

Marius Arvinte and Jon Tamir have been supported by NSF IFML 2019844 award, ONR grant N00014-19-1-2590, NIH Grant U24EB029240, and an AWS Machine Learning Research Award.

We thank the anonymous NeurIPS reviewers for their helpful and considerate feedback.

Finally, we would like to thank the experts who graciously helped with our image assessment study.

# 8 Funding Transparency Statements

## 8.1 Funding (financial activities supporting the submitted work):

Funding in direct support of this work: NSF Grants CCF 1763702, 1934932, AF 1901281, 2008710, 2019844, 1751040, 2008868, the NSF IFML 2019844 award, ONR grant N00014-19-1-2590, NIH Grant U24EB029240, and an AWS Machine Learning Research Award, as well as research gifts by Western Digital, Interdigital, WNCG and MLL, computing resources from TACC and the Archie Straiton Fellowship.

## 8.2 Competing Interests (financial activities outside the submitted work):

Additional revenues related to this work: Internship at Intel and Google.

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
