

(a) Brains - Vertical Mask

(b) Brains - Horizontal mask

(c) Abdomens

(d) Proton Density Knees

Figure 5: Average test SSIM in various scenarios, across a range of acceleration factors $R$. Higher $R$ indicates a smaller number of acquired measurements. Our approach mostly shows the best performance and lowest reconstruction variance both in- and out-of-distribution at test-time. Shaded regions indicate 95% confidence intervals. Note that we trained baselines on MVUE images and hence these numerical values should not be compared with those in literature trained on RSS images (see Appendix A.1 for a more detailed discussion).

# A  Appendix: Additional Metrics

Figure 5 shows the test SSIM evaluated in the same conditions as Figure 2 in the main text. This highlights that our model is also robust in this metric.

We observe that our method has significant noise in the background. Hence, we also report the masked SSIM and PSNR values in Figures 6 and 7. The mask zeros out all coordinates whose absolute value is smaller than 0.05 times the maximum absolute value in the fully-sampled MVUE.

## A.1  MVUE vs. RSS

The difference in numerical values between our results and the publicly available fastMRI leaderboard, as well as original results in the published baseline papers baselines comes from training and evaluating all methods on MVUE instead of RSS images. This is a design choice that we have made for all baselines, since our goal is to compare with a wide range of previous methods in a fair way.

Algorithms that output a complex-valued image (such as ours and L1-Wavelet) as a solution to the optimization in Eqn (2) will artificially perform worse (w.r.t. E2E methods) when compared to the RSS ground truth, even when the output is of similar or higher quality, due to the bias in the RSS. Since there is no way to obtain a good RSS score with these algorithms, this justifies our choice to train and evaluate all methods on MVUE.

To the best of our knowledge, a rigorous, reproducible comparison between end-to-end models trained on RSS or MVUE images has not been made in prior work. The recent work of [33] has also discussed this point. To illustrate our claim of incompatibility between the two estimates, as well as the importance of qualitative inspection, we provide two simple, easy-to-verify examples.

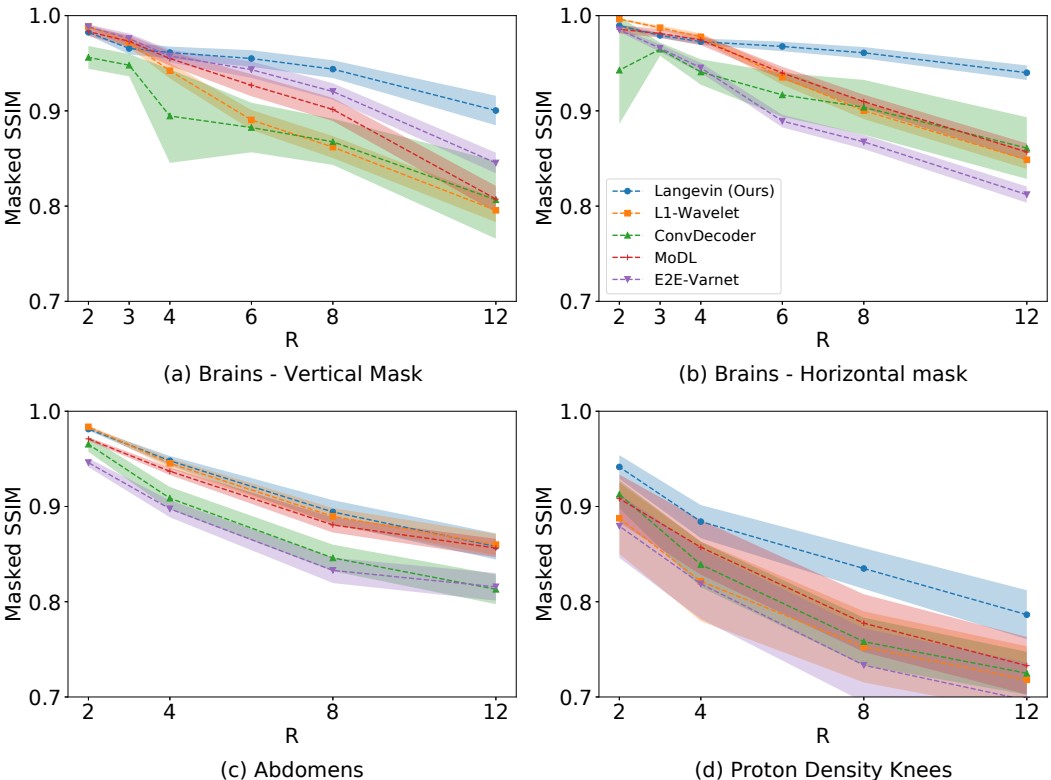

Figure 6: Average test SSIM, with masking, in various scenarios across a range of acceleration factors $R$. The mask zeros out all coordinates whose absolute value is smaller than 0.05 times the maximum absolute value in the fully-sampled MVUE, and this reduces the effect of noise in the background. Higher $R$ indicates a smaller number of acquired measurements. Our approach mostly shows the best performance and lowest reconstruction variance both in- and out-of-distribution at test-time. Shaded regions indicate 95% confidence intervals. Note that we trained baselines on MVUE images and hence these numerical values should not be compared with those in literature trained on RSS images (see Appendix A.1 for a more detailed discussion).

1. We compare the fully sampled MVUE reconstruction (with ESPiRIT estimated maps) with the fully sampled RSS reconstruction, on T2 brain scans: we find that the SSIM is slightly larger than $0.8$. This is a large penalty (as per Fig. 1), even though the two images are virtually indistinguishable and known to be clinically equivalent (see discussions of SENSE vs. GRAPPA in [33]). This would unfairly penalize the family of methods that explicitly solve the inverse problem. Since E2E methods can be trained to target the MVUE directly, this justifies our choice for using the MVUE as the reference image.

2. We point to the public knee fastMRI leaderboard at https://fastmri.org/leaderboards. Selecting "Multi-coil Knee" and "4x" acceleration, we inspect the two following submissions:

   - "zero-filling", which does zero-filling RSS reconstruction, has an SSIM of $0.804$ and considerable artifacts.
   - "Baseline Classical Reconstruction Model", which applies compressed sensing with the ESPiRIT algorithm, has a much poorer SSIM score of $0.6275$, but produces qualitatively superior reconstructions.

# B   Appendix: Theory

**Lemma B.1** ($\mathcal{W}_q$ implies $(\delta, \alpha)$-$\mathcal{W}_\infty$). *If two distributions $\mu$ and $\nu$ satisfy $\mathcal{W}_q(\mu, \nu) \leq \varepsilon$ for some $q \geq 1$, then they satisfy $(\delta, \delta)$-$\mathcal{W}_\infty(\mu, \nu) \leq \varepsilon/\delta^{1/q}$. Futhermore, there exist distributions that satisfy $(\delta, \delta)$-$\mathcal{W}_\infty(\mu, \nu) \leq \varepsilon$, but $\mathcal{W}_q(\mu, \nu) = \infty$ for all $q \geq 1$.*

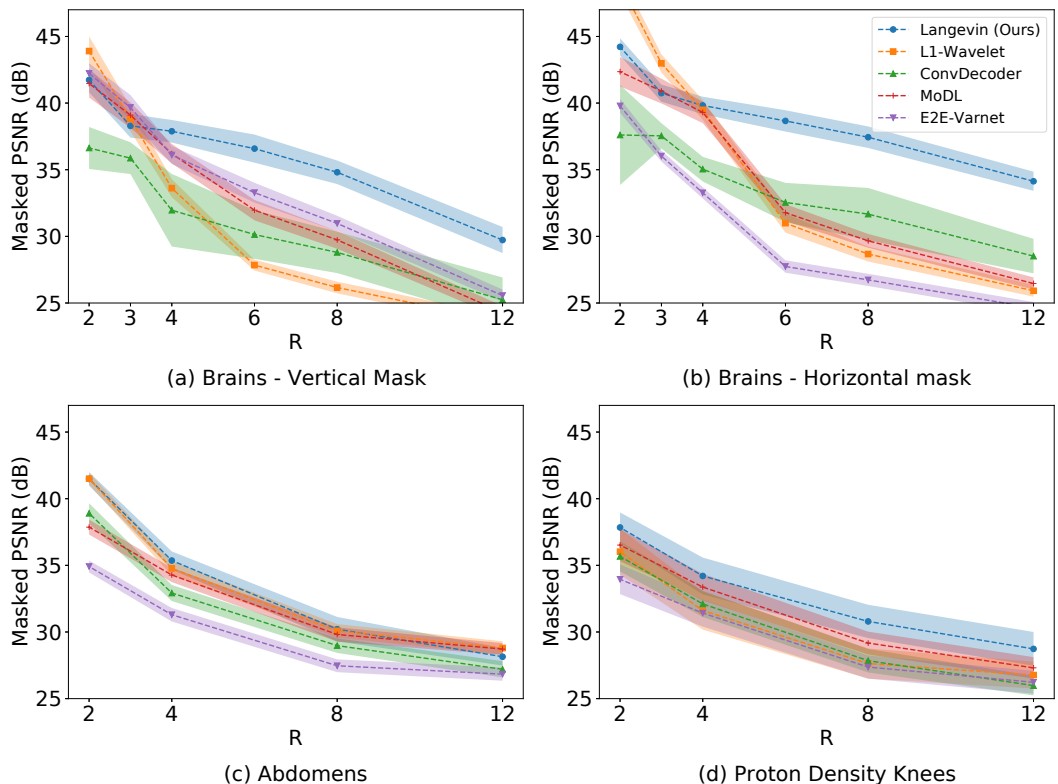

Figure 7: Average test PSNR, with masking, in various scenarios across a range of acceleration factors $R$. The mask zeros out all coordinates whose absolute value is smaller than 0.05 times the maximum absolute value in the fully-sampled MVUE, and this reduces the effect of noise in the background. Higher $R$ indicates a smaller number of acquired measurements. Our approach mostly shows the best performance and lowest reconstruction variance both in- and out-of-distribution at test-time. Shaded regions indicate 95% confidence intervals. Note that we trained baselines on MVUE images and hence these numerical values should not be compared with those in literature trained on RSS images (see Appendix A.1 for a more detailed discussion).

*Proof.* Let $\Gamma$ be a coupling between $\mu, \nu$ such that $\mathbb{E}_{(u,v)\sim\Gamma}\left[\|u-v\|^q\right] \leq \varepsilon^q$. Then an application of Markov's inequality gives

$$\Pr[\|u-v\| \geq \varepsilon/\delta^{1/q}] \leq \delta. \tag{5}$$

Now, we can split the distribution $\Gamma$ into two unnormalized components $\Gamma', \Gamma''$ defined as

$$\Gamma'(u,v) = \Gamma(u,v)\mathbf{1}\{\|u-v\| < \varepsilon/\delta^{1/q}\},$$
$$\Gamma''(u,v) = \Gamma(u,v)\mathbf{1}\{\|u-v\| \geq \varepsilon/\delta^{1/q}\}.$$

Using $\Gamma', \Gamma''$, we can define measures $\mu', \mu'', \nu', \nu''$, via

$$\mu'(B) := \Gamma'(B, \Omega),$$
$$\mu''(B) := \Gamma''(B, \Omega),$$
$$\nu'(B) := \Gamma'(\Omega, B),$$
$$\nu''(B) := \Gamma''(\Omega, B),$$

where $B$ is any measurable set and $\Omega$ is the state-space.

Since $\Gamma$ is a valid coupling between $\mu, \nu$, and $\Gamma', \Gamma''$ are disjoint distributions, for any measurable $B \subseteq \Omega$, we have:

$$\mu(B) = \Gamma(B, \Omega),$$
$$= \Gamma'(B, \Omega) + \Gamma''(B, \Omega),$$
$$= \mu'(B) + \mu(B''),$$
$$= \mu'(\Omega)\frac{\mu'(B)}{\mu'(\Omega)} + \mu''(\Omega)\frac{\mu''(B)}{\mu''(\Omega)}.$$

Using Eqn (5), we can conclude that $\mu'(\Omega) \geq 1 - \delta, \mu''(\Omega) \leq \delta$. Setting $\mu' \leftarrow \mu'/\mu'(\Omega)$ and $\mu'' \leftarrow \mu''/\mu''(\Omega)$, we can now rewrite $\mu$ as $\mu = (1-\delta)\mu' + \delta\mu''$. A similar argument for $\nu$ gives $\nu = (1-\delta)\nu' + \delta\nu''$.

By construction, $\mu', \nu'$ can be $\mathcal{W}_\infty$ coupled via $\Gamma'$ to within a distance of $\varepsilon/\delta^{1/q}$. This shows that $(\delta, \delta)\text{-}\mathcal{W}_\infty(\mu, \nu) \leq \varepsilon/\delta^{1/q}$.

Now we need to show that two distributions can be close in $(\delta, \delta)\text{-}\mathcal{W}_\infty$, but $\mathcal{W}_q = \infty$ for all $q$. Consider two scalar distributions $\mu, \nu$ defined as

$$\mu = \begin{cases} 0 & \text{with probability } 1 - \delta, \\ r & \text{with probability } \delta, \end{cases},$$

$$\nu = \begin{cases} \varepsilon & \text{with probability } 1 - \delta, \\ -r & \text{with probability } \delta. \end{cases}$$

Clearly, these distributions satisfy $(\delta, \delta)\text{-}\mathcal{W}_\infty(\mu, \nu) \leq \varepsilon$, but $\mathcal{W}_q(\mu, \nu) \approx r$ for all $q$. As $r \to \infty$, we get $\mathcal{W}_q(\mu, \nu) \to \infty$ for all $q \geq 1$.

$\square$

## B.1 Proof of Theorem 3.3

In order to prove the Theorem, we make use of the following three Lemmas from [45].

**Lemma B.2.** *[45] For $c \in [0, 1]$, let $H := (1-c)H_0 + cH_1$ be a mixture of two absolutely continuous distributions $H_0, H_1$ admitting densities $h_0, h_1$. Let $y$ be a sample from the distribution $H$, such that $y|z^* \sim H_{z^*}$ where $z^* \sim Bernoulli(c)$.*

*Define $\widehat{c}_y = \frac{ch_1(y)}{(1-c)h_0(y)+ch_1(y)}$, and let $\widehat{z}|y \sim Bernoulli(\widehat{c}_y)$ be the posterior sampling of $z^*$ given $y$. Then we have*

$$\Pr_{z^*, y, \widehat{z}}[z^* = 0, \widehat{z} = 1] \leq 1 - TV(H_0, H_1).$$

**Lemma B.3.** *[45] Let $y$ be generated from $x^*$ by a Gaussian measurement process with noise rate $\sigma$. For a fixed $\tilde{x} \in \mathbb{R}^n$, and parameters $\eta > 0, c \geq 4e^2$, let $P_{out}$ be a distribution supported on the set*

$$S_{\tilde{x}, out} := \{x \in \mathbb{R}^n : \|x - \tilde{x}\| \geq c(\eta + \sigma)\}.$$

*Let $P_{\tilde{x}}$ be a distribution which is supported within an $\eta-$radius ball centered at $\tilde{x}$.*

*For a fixed $A$, let $H_{\tilde{x}}$ denote the distribution of $y$ when $x^* \sim P_{\tilde{x}}$. Let $H_{out}$ denote the corresponding distribution of $y$ when $x^* \sim P_{out}$. Then we have:*

$$\mathbb{E}_{A}[TV(H_{\tilde{x}}, H_{out})] \geq 1 - 4e^{-\frac{m}{2}\log\left(\frac{c}{4e^2}\right)}.$$

**Lemma B.4.** *[45] Let $R, P$, denote arbitrary distributions over $\mathbb{R}^n$ such that $\mathcal{W}_\infty(R, P) \leq \varepsilon$.*

*Let $x^* \sim R$ and $z^* \sim P$ and let $y$ and $u$ be generated from $x^*$ and $z^*$ via a Gaussian measurement process with $m$ measurements and noise rate $\sigma$. Let $\widehat{x} \sim P(\cdot|y, A)$ and $\widehat{z} \sim P(\cdot|u, A)$. For any $d > 0$, we have*

$$\Pr_{x^*, A, w, \widehat{x}}[\|x^* - \widehat{x}\| \geq d + \varepsilon] \leq e^{-\Omega(m)} + e^{\left(\frac{4\varepsilon(\varepsilon+2\sigma)m}{2\sigma^2}\right)} \Pr_{z^*, A, w, \widehat{z}}[\|z^* - \widehat{z}\| \geq d].$$

**Theorem 3.3.** *Let $\delta, \alpha \in [0,1]$, and $\varepsilon > 0$ be parameters. Let $\mu, \nu$ be arbitrary distributions over $\mathbb{R}^N$ satisfying $(\delta, \alpha)$-$\mathcal{W}_\infty(\mu, \nu) \leq \varepsilon$. Let $x^* \sim \mu$ and suppose $y = Ax^* + w$, where $A \in \mathbb{R}^{M \times N}$ and $w \in \mathbb{R}^M$ are i.i.d. Gaussian normalized such that $A_{ij} \sim \mathcal{N}(0, 1/M)$ and $w_i \sim \mathcal{N}(0, \sigma^2/M)$, with $\sigma \gtrsim \varepsilon$. Given $y$ and the fixed matrix $A$, let $\widehat{x}$ be the output of posterior sampling with respect to $\nu$.*

*Then for $M \geq O\left(\log\left(\frac{1}{1-\alpha}\right) + \min(\log \operatorname{Cov}_{\sigma,\delta}(\mu), \log \operatorname{Cov}_{\sigma,\delta}(\nu))\right)$, there exists a universal constant $c > 0$ such that with probability at least $1 - e^{-\Omega(M)}$ over $A, w$,*

$$\Pr_{x^* \sim \mu, \widehat{x} \sim \nu(\cdot | y)} [\|x^* - \widehat{x}\| \geq c(\varepsilon + \sigma)] \leq \delta + e^{-\Omega(M)}.$$

*Proof.* We know from $(\delta, \alpha)$-$\mathcal{W}_\infty(\mu, \nu) \leq \varepsilon$ that there exist $\mu', \nu', \mu'', \nu''$ and a finite distribution $Q$ supported on a set $S$ such that

1. $\mathcal{W}_\infty(\mu', \nu') \leq \varepsilon$,

2. $\min\{\mathcal{W}_\infty(\nu', Q), \mathcal{W}_\infty(\mu', Q)\} \leq \sigma$,

3. $\mu = (1-\delta)\mu' + \delta\mu''$ and $\nu = (1-\alpha)\nu' + \alpha\nu''$.

Suppose $\mathcal{W}_\infty(\nu', Q) \leq \sigma$. If not, then $\mathcal{W}_\infty(\mu', Q) \leq \sigma$, and by (1), we see that $\mathcal{W}_\infty(\nu', Q) \leq \sigma + \varepsilon$, and we will use this in the proof instead. By decomposing $\mu = (1-\delta)\mu' + \delta\mu''$, we have

$$\Pr_{x^* \sim \mu, \widehat{x} \sim \nu(\cdot|y)} [\|x^* - \widehat{x}\| \geq (2c+1)\sigma + \varepsilon] \leq \delta + (1-\delta) \Pr_{x^* \sim \mu', \widehat{x} \sim \nu(\cdot|y)} [\|x^* - \widehat{x}\| \geq (2c+1)\sigma + \varepsilon].$$
(6)

We now bound the second term on the right hand side of the above equation. For this term, consider the joint distribution over $x^*, A, w, \widehat{x}$. By Lemma B.4, we can replace $x^* \sim \mu'$ with $z^* \sim \nu'$, replace $y = Ax^* + w$ with $u = Az^* + w$, and replace $\widehat{x} \sim \nu(\cdot|A, y)$ with $\widehat{z} \sim \nu(\cdot|A, u)$ to get the following bound

$$\Pr_{x^* \sim \mu', A, w, \widehat{x} \sim \nu(\cdot|A,y)} [\|x^* - \widehat{x}\| \geq (2c+1)\sigma + \varepsilon] \leq e^{-\Omega(m)} + e^{\left(\frac{2\varepsilon(\varepsilon+2\sigma)m}{\sigma^2}\right)} \Pr_{z^* \sim \nu', A, w, \widehat{z} \sim \nu(\cdot|u,A)} [\|z^* - \widehat{z}\| \geq (2c+1)\sigma].$$
(7)

We now bound the second term in the right hand side of the above inequality. Let $\Gamma$ denote an optimal $\mathcal{W}_\infty$−coupling between $\nu'$ and $Q$.

For each $\tilde{z} \in S$, the conditional coupling can be defined as

$$\Gamma(\cdot|\tilde{z}) = \frac{\Gamma(\cdot, \tilde{z})}{Q(\tilde{z})}.$$

By the $\mathcal{W}_\infty$ condition, each $\Gamma(\cdot|\tilde{z})$ is supported on a ball of radius $\sigma$ around $\tilde{z}$.

Let $E = \{z^*, \widehat{z} \in \mathbb{R}^n : \|z^* - \widehat{z}\| \geq (2c+1)\sigma\}$ denote the event that $z^*, \widehat{z}$ are far apart. By the coupling, we can express $\nu'$ as

$$\nu' = \sum_{\tilde{z} \in S} Q(\tilde{z})\Gamma(\cdot|\tilde{z}).$$

This gives

$$\Pr_{z^* \sim \nu', A, w, \widehat{z} \sim \nu(\cdot|A,u)} [E] = \sum_{\tilde{z}^* \in S} Q(\tilde{z}^*) \mathbb{E}_{z^* \sim \Gamma(\cdot|\tilde{z}^*), A, w, \widehat{z} \sim \nu(\cdot|A,u)} [1_E].$$

For each $\tilde{z}^* \in S$, we now bound $Q(\tilde{z}^*) \mathbb{E}_{z^* \sim \Gamma(\cdot|\tilde{z}^*), A, w, \widehat{z} \sim \nu(\cdot|A,u)} [1_E]$.

For each $\tilde{z}^* \in S$, we can write $\nu$ as $\nu = (1-\alpha)Q_{\tilde{z}^*}\nu_{\tilde{z}^*,0} + c_{\tilde{z}^*,1}\nu_{\tilde{z}^*,1} + c_{\tilde{z}^*,2}\nu_{\tilde{z}^*,2}$, where the components of the mixture are defined in the following way. The first component $\nu_{\tilde{z}^*,0}$ is $\Gamma(\cdot|\tilde{z}^*)$, the second component is supported within a $2c\sigma$ radius of $\tilde{z}^*$, and the third component is supported outside a $2c\sigma$ radius of $\tilde{z}^*$.

Formally, let $B_{\tilde{z}^*}$ denote the ball of radius $c\sigma$ centered at $\tilde{z}^*$, and let $B_{\tilde{z}^*}^c$ be its complement. The constants are defined via the following Lebesque integrals, and the mixture components for any Borel measurable $B$ are defined as

$$c_{\tilde{z}^*,1} := \int_{B_{\tilde{z}^*}} d\nu - (1-\alpha)\, Q_{\tilde{z}^*} \int_{B_{\tilde{z}^*}} d\Gamma(\cdot|\tilde{z}^*),$$

$$c_{\tilde{z}^*,2} := \int_{B_{\tilde{z}^*}^c} d\nu - (1-\alpha)\, Q_{\tilde{z}^*} \int_{B_{\tilde{z}^*}^c} d\Gamma(\cdot|\tilde{z}^*),$$

$$\nu_{\tilde{z}^*,0}(B) := \Gamma(B \cap B_{\tilde{z}^*}|\tilde{z}^*) = \Gamma(B|\tilde{z}^*) \text{ since } \mathrm{supp}(\Gamma(\cdot|\tilde{z}^*)) \subset B_{\tilde{z}^*},$$

$$\nu_{\tilde{z}^*,1}(B) := \begin{cases} \frac{1}{c_{\tilde{z}^*,1}}\nu(B \cap B_{\tilde{z}^*}) - \frac{1-\alpha}{c_{\tilde{z}^*,1}}Q_{\tilde{z}^*}\Gamma(B \cap B_{\tilde{z}^*}|\tilde{z}^*) & \text{if } c_{\tilde{z}^*,1} > 0, \\ \text{do not care} & \text{otherwise.} \end{cases},$$

$$\nu_{\tilde{z}^*,2}(B) := \begin{cases} \frac{1}{c_{\tilde{z}^*,2}}\nu(B \cap B_{\tilde{z}^*}^c) - \frac{1-\alpha}{c_{\tilde{z}^*,2}}Q_{\tilde{z}^*}\Gamma(B \cap B_{\tilde{z}^*}^c|\tilde{z}^*) & \text{if } c_{\tilde{z}^*,2} > 0, \\ \text{do not care} & \text{otherwise.} \end{cases}.$$

Notice that if $z^*$ is sampled from $\Gamma(\cdot|\tilde{z}^*)$, then by the $W_\infty$ condition, we have $\|z^* - \tilde{z}^*\| \le \sigma$. Furthermore, if $\hat{z}$ is $(2c+1)\sigma$ far from $z^*$, an application of the triangle inequality implies that it must be distributed according to $\nu_{\tilde{z}^*,2}$. That is,

$$Q(\tilde{z}^*) \underset{z^*\sim\Gamma(\cdot|\tilde{z}^*),A,w,\hat{z}\sim\nu(\cdot|A,u)}{\mathbb{E}} [1_E] \le \underset{A,w,z^*}{\mathbb{E}} \Pr\left[z^* \sim \nu_{\tilde{z}^*,0}, \hat{z} \sim \nu_{\tilde{z}^*,2}(\cdot|u)\right]$$

$$\le \frac{1}{1-\alpha} \underset{A}{\mathbb{E}}\left[1 - TV(H_{\tilde{z}^*,0}, H_{\tilde{z}^*,2})\right],$$

where $H_{\tilde{z}^*,0}, H_{\tilde{z}^*,2}$ are the push-forwards of $\nu_{\tilde{z}^*,0},\nu_{\tilde{z}^*,2}$ for $A$ fixed and the last inequality follows from Lemma B.2.

Notice that if we sum over all $\tilde{z}^* \in S$, then the LHS of the above inequality is an expectation over $z^* \sim \nu'$. This gives:

$$\underset{z^*\sim\nu',A,w,\hat{z}\sim\nu(\cdot|u,A)}{\Pr} [E] \le \frac{1}{1-\alpha} \sum_{\tilde{z}^*\in S} \underset{A}{\mathbb{E}}\left[1 - TV(H_{\tilde{z}^*,0}, H_{\tilde{z}^*,2})\right].$$

Notice that $\nu_{\tilde{z}^*,0}$ is supported within an $\sigma-$ball around $\tilde{z}^*$, and $\nu_{\tilde{z}^*,2}$ is supported outside a $2c\sigma-$ball of $\tilde{z}^*$. By Lemma B.3 we have

$$\underset{A}{\mathbb{E}}[TV(H_{\tilde{z}^*,0}, H_{\tilde{z}^*,2})] \ge 1 - 4e^{-\frac{m}{2}\log\left(\frac{c}{4e^2}\right)}.$$

This implies

$$\underset{z^*\sim\nu',A,w,\hat{z}\sim\nu(\cdot|u,A)}{\Pr} [\|z^* - \hat{z}\| \ge (2c+1)\sigma] \le \frac{1}{1-\alpha} \sum_{\tilde{z}^*\in S} \underset{A}{\mathbb{E}}\left[(1 - TV(H_{\tilde{z}^*,0}, H_{\tilde{z}^*,2}))\right],$$

$$\le \frac{1}{1-\alpha} 4|S| e^{-\frac{m}{2}\log\left(\frac{c}{4e^2}\right)},$$

$$\le 4e^{-\frac{m}{4}\log\left(\frac{c}{4e^2}\right)},$$

where the last inequality is satisfied if $m \ge 4\log\left(\frac{1}{1-\alpha}\right) + 4\log(|S|)$.

Substituting in Eqn (7), if $c > 4\exp\left(2 + \frac{8\varepsilon(\varepsilon+2\sigma)}{\sigma^2}\right)$, we have

$$\underset{x^*\sim\mu',A,w,\hat{x}\sim\nu(\cdot|A,y)}{\Pr} [\|x^* - \hat{x}\| \ge (2c+1)\sigma + \varepsilon] \le e^{-\Omega(m)}.$$

This implies that there exists a set $S_{A,w}$ over $A, w$ satisfying $\Pr_{A,w}[S_{A,w}] \geq 1 - e^{-\Omega(m)}$, such that for all $A, w \in S_{A,w}$, we have

$$\Pr_{x^* \sim \mu', \widehat{x} \sim \nu(\cdot|y)} [\|x^* - \widehat{x}\| \geq (2c+1)\sigma + \varepsilon] \leq e^{-\Omega(m)}.$$

Substituting in Eqn (6), we have

$$\Pr_{x^* \sim \mu, \widehat{x} \sim \nu(\cdot|y)} [\|x^* - \widehat{x}\| \geq (2c+1)\sigma + \varepsilon] \leq \delta + e^{-\Omega(m)}.$$

Rescaling $c$ gives us our result.

At the beginning of the proof, we had assumed that $\mathcal{W}_\infty(\nu', Q) \leq \sigma$. If instead $\mathcal{W}_\infty(\mu', Q) \leq \sigma$, then we need to replace $\sigma$ in the above bound by $\sigma + \varepsilon$. Rescaling $c$ in the above bound gives us the Theorem statement.

$\square$

## B.2  Proof of Theorem 3.4

**Theorem 3.4.** *Let $d(\cdot, \cdot)$ be an arbitrary metric over $\mathbb{R}^N \times \mathbb{R}^N$. Let $x^* \sim \mu$ and let $y = \mathcal{A}(x^*)$ be measurements generated from $x^*$ for some arbitrary forward operator $\mathcal{A} : \mathbb{R}^N \to \mathbb{R}^M$. Then if there exists an algorithm that uses $y$ as inputs and outputs $x'$ such that*

$$d(x^*, x') \leq \varepsilon \text{ with probability } 1 - \delta,$$

*then posterior sampling $\widehat{x} \sim \mu(\cdot|y)$ will satisfy*

$$d(x^*, \widehat{x}) \leq 2\varepsilon \text{ with probability } \geq 1 - 2\delta.$$

*Proof.* By the statement of the Lemma, and conditioning on the measurements $y$, we have

$$1 - \delta = \Pr[d(x^*, x') \leq \varepsilon] = \mathbb{E}_y \left( \Pr[d(x^*, x') \leq \varepsilon | y] \right).$$

Using a similar conditioning for the event $d(x^*, \widehat{x}) \leq 2\varepsilon$, we get

$$\begin{aligned}
\Pr[d(x^*, \widehat{x}) \leq 2\varepsilon] &= \mathbb{E}_y \left( \Pr[d(x^*, \widehat{x}) \leq 2\varepsilon | y] \right), \\
&\geq \mathbb{E}_y \left( \Pr[d(x^*, x') \leq \varepsilon \wedge d(x', \widehat{x}) \leq \varepsilon | y] \right), \\
&= \mathbb{E}_y \left( \Pr[d(x^*, x') \leq \varepsilon | y] \cdot \Pr[d(x', \widehat{x}) \leq \varepsilon | y] \right), \\
&= \mathbb{E}_y \left( \Pr[d(x^*, x') \leq \varepsilon | y]^2 \right), \\
&\geq \left( \mathbb{E}_y \left( \Pr[d(x^*, x') \leq \varepsilon | y] \right) \right)^2, \\
&= (1 - \delta)^2 \geq 1 - 2\delta,
\end{aligned}$$

where the second line follows from a triangle inequality, the third line follows since $x^*, \widehat{x}$ are independent conditioned on $y$, the fourth line follows since $\widehat{x}|y$ is distributed according to $x^*|y$, and the fifth line follows from Jensen's inequality.

$\square$

# C  Appendix: fastMRI Brain

## C.1  Examples of Sampling Masks

Figure 8 shows example of some of the masks used throughout the experiments in the paper and their corresponding reconstructions. Note that the type of mask used is coupled with the scan parameters (e.g., two-dimensional slices from a three-dimensional scan will use a 2D grid of points).

We also highlight that, in all cases, a central region of the k-space is kept fully sampled and is used to estimate the coil sensitivity maps for all methods. The bottom row of Figure 8 shows naive reconstructions of a single coil image using the zero-filled k-space. This shows that different types of masks lead to different types of aliasing patterns in the image domain, motivating the need for robust image reconstruction algorithms.

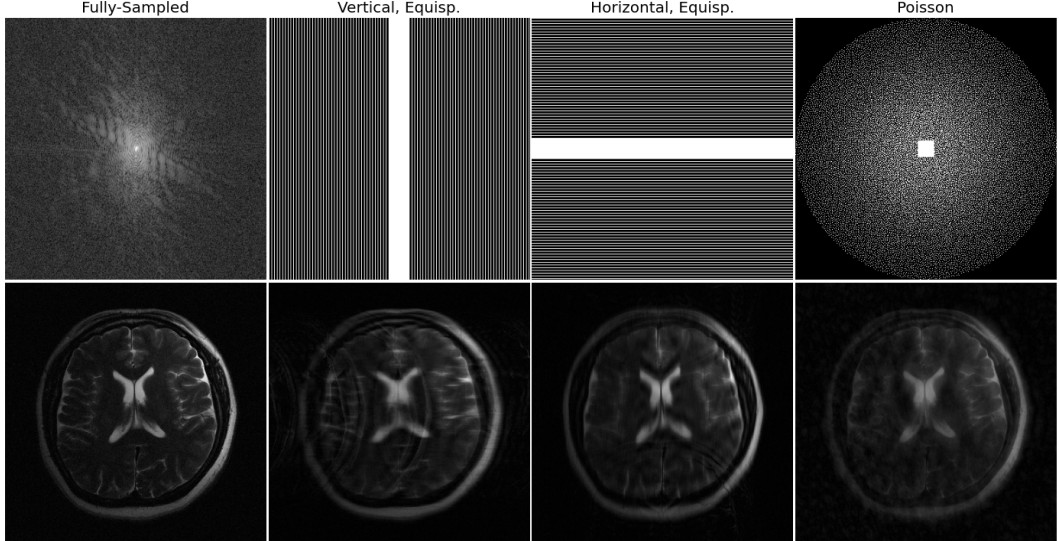

Figure 8: Examples of sampling patterns used throughout the experiments (top) and naive reconstructions (bottom). Top: The leftmost image shows the log-magnitude of the fully sampled k-space measurements corresponding to a single coil. The remaining images show three possible sampling masks, all with acceleration factor $R = 4$ but drastically different patterns. Bottom: Each image shows the magnitude of the reconstruction obtained by a two-dimensional IFFT applied to the sampled k-space.

## C.2 More Exemplar Reconstructions

Figures 9 throughout 14 show detailed qualitative reconstructions on different brain scans from the fastMRI dataset. We highlight Figures 13 and 14, which represent a contrast shift from the in-distribution data (T1 and FLAIR vs. T2, respectively). Our method still produces excellent qualitative reconstructions.

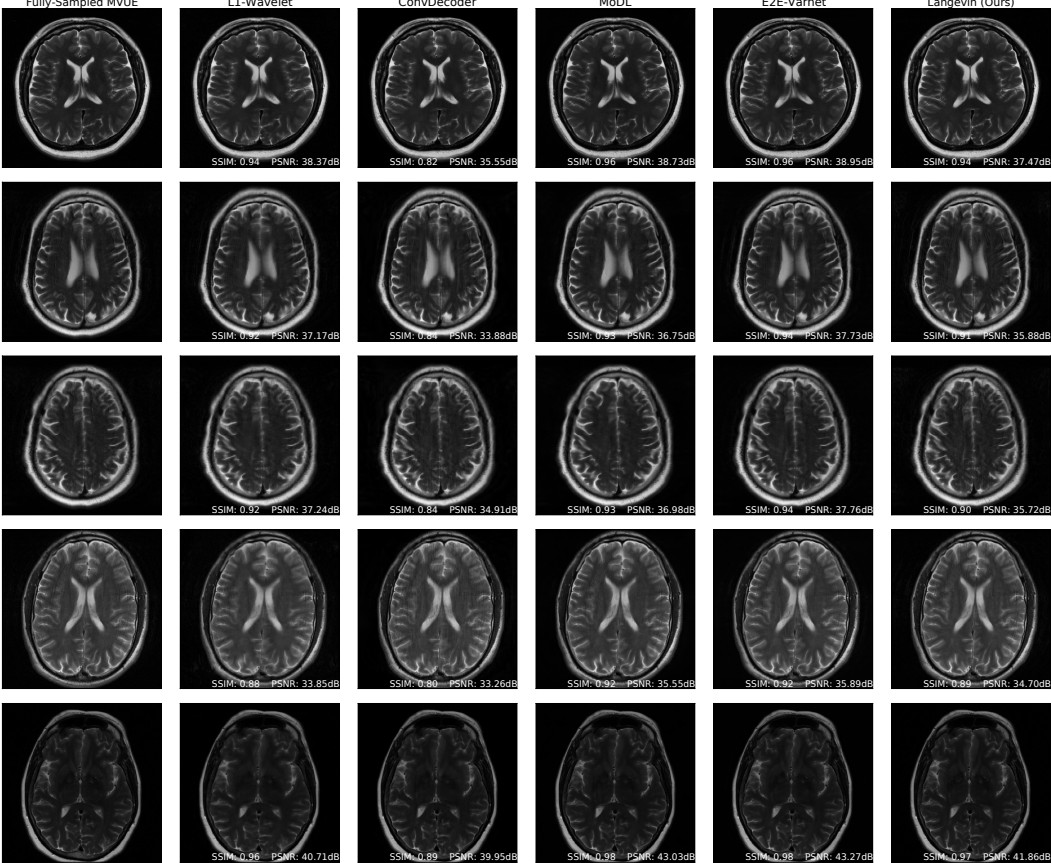

Figure 9: In-distribution brain reconstructions, at an acceleration factor of $R = 3$ and an equispaced vertical mask in k-space. Our model was trained on T2-weighted brain images from the fastMRI dataset. These results show that our method is competitive with state-of-the-art methods such as E2E-VarNet.

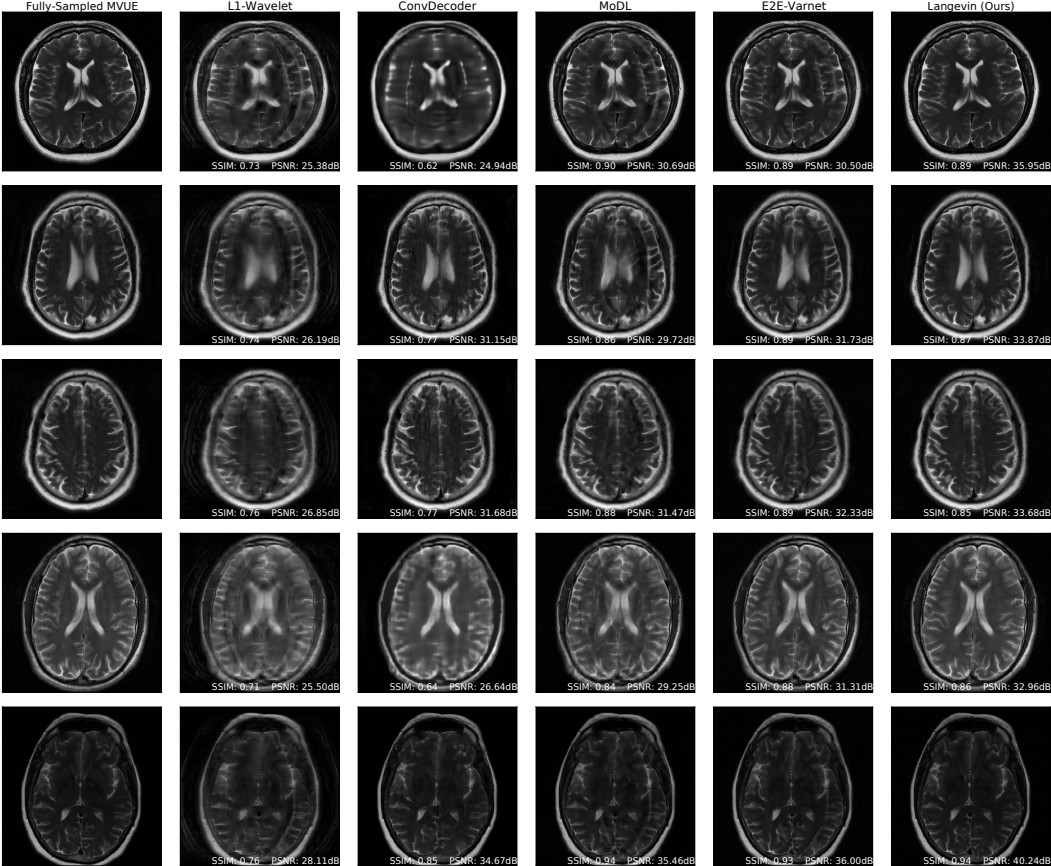

Figure 10: In-distribution brain reconstructions, at an acceleration factor of $R = 6$ and an equispaced vertical mask in k-space. Our model was trained on T2-weighted brain images from the fastMRI dataset. These results show that our method retains its performance at higher acceleration factors.

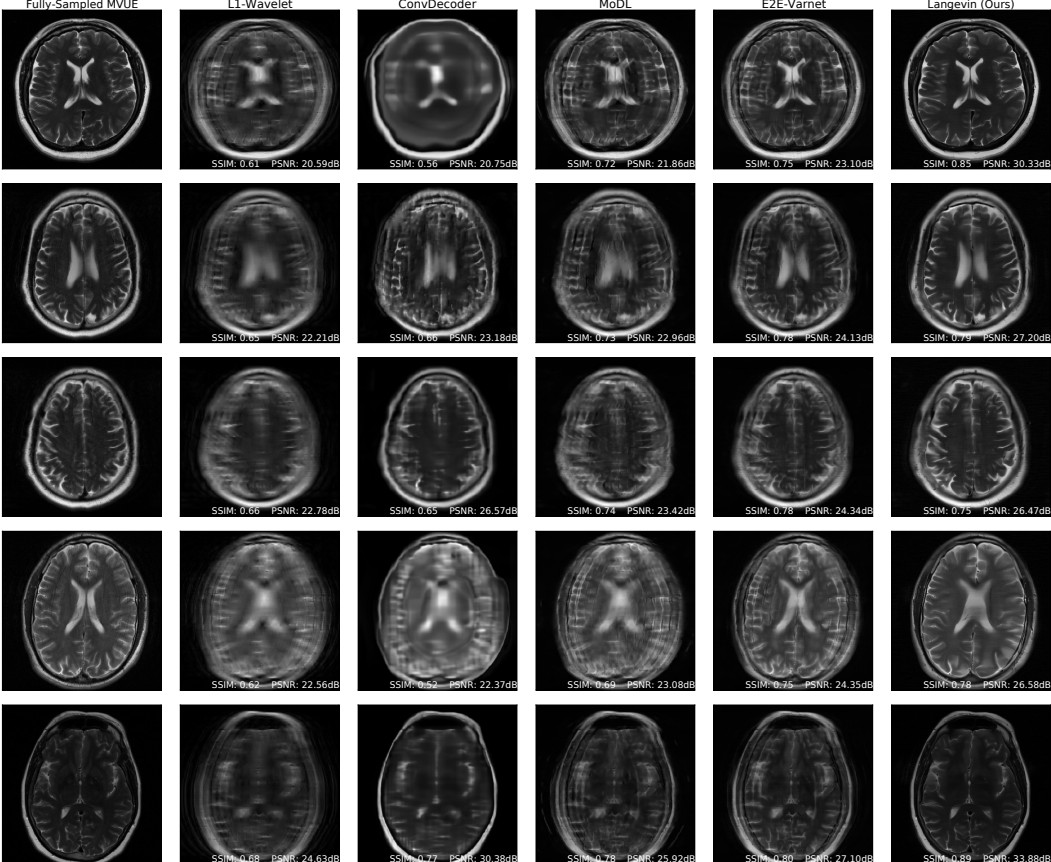

Figure 11: Brain reconstructions, at an acceleration factor of $R = 12$ and an equispaced vertical mask in k-space. Our model was trained on T2-weighted brain images from the fastMRI dataset. These results show that our method has significantly fewer artifacts than baselines.

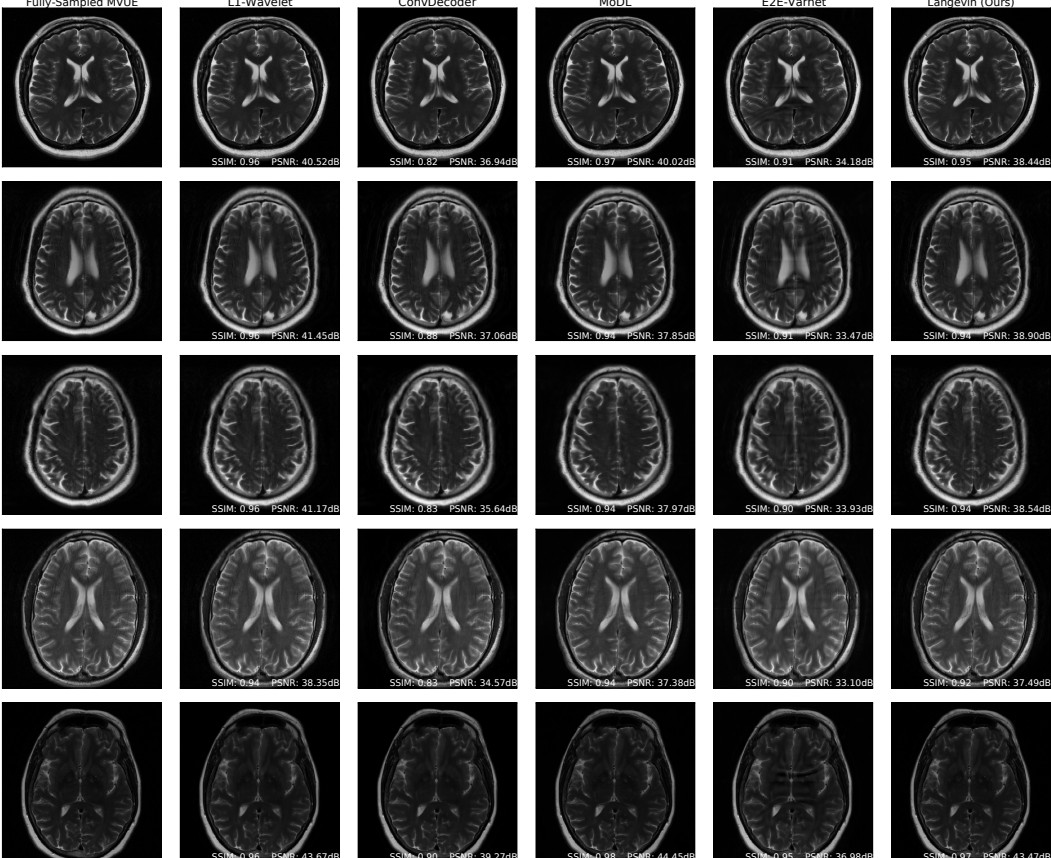

Figure 12: Brain reconstructions under a mask shift, at an acceleration of $R = 3$. MoDL and E2E-VarNet were trained using an equispaced vertical mask, while these experiments were run using an equispaced *horizontal* mask. Our method is robust to the mask shift, as our generative prior was trained without any knowledge of the measurement process. ConvDecoder and L1-Wavelets are untrained methods, and hence are robust to the mask shift.

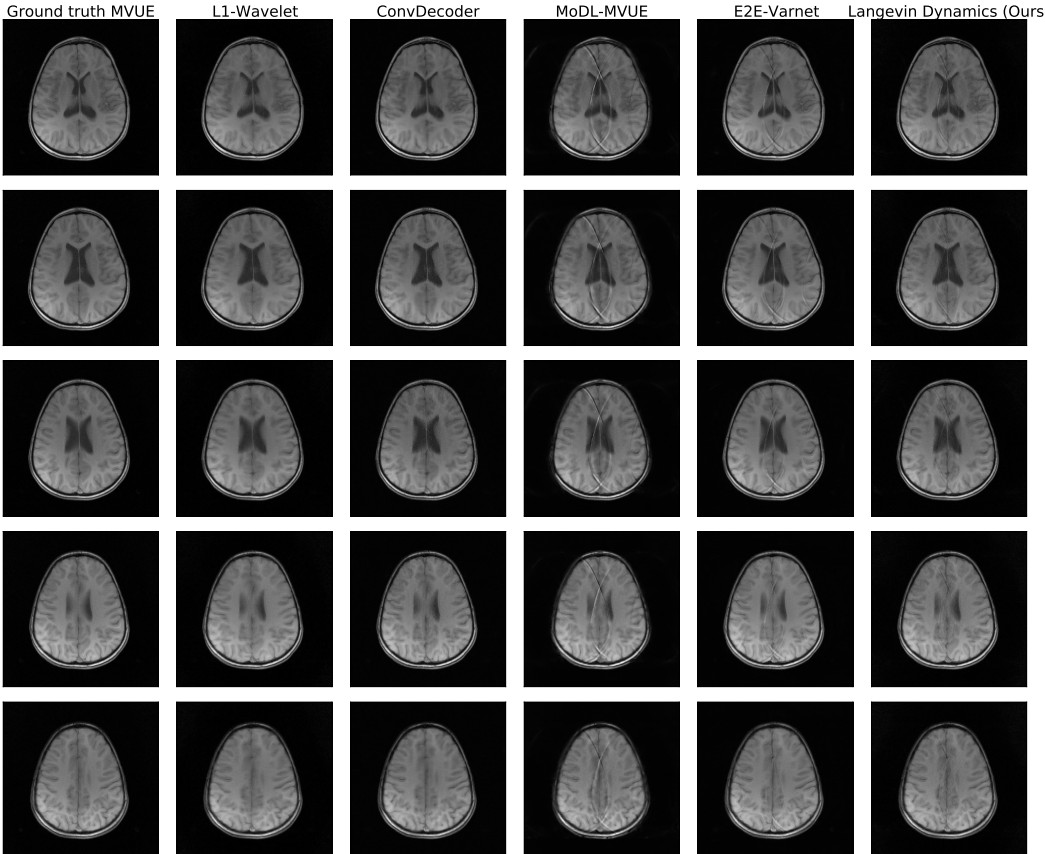

Figure 13: Brain reconstructions under a contrast shift, at an acceleration of $R = 4$. Our method was trained on T2-weighted brains, while these are T1-weighted brains, and our method is clearly robust to this contrast shift.

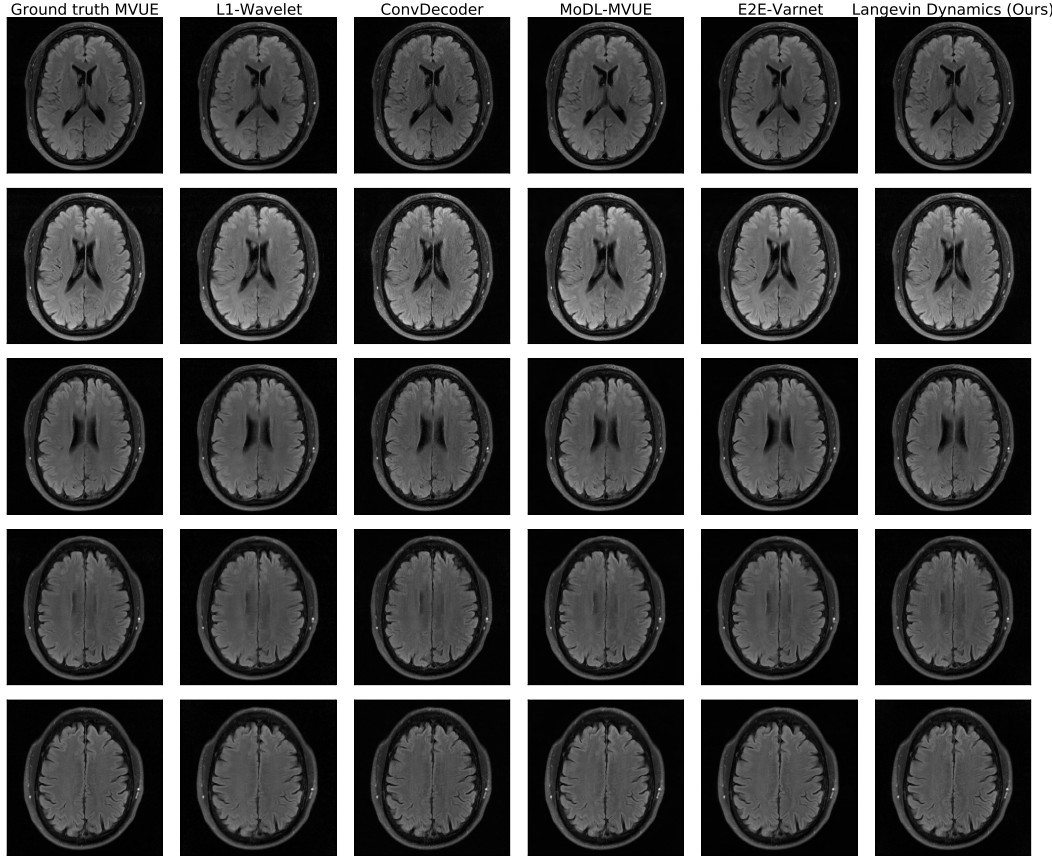

Figure 14: Brain reconstructions under a contrast shift, at an acceleration of $R = 4$. Our method was trained on T2-weighted brains, while these are FLAIR brains, and our method is clearly robust to this contrast shift.

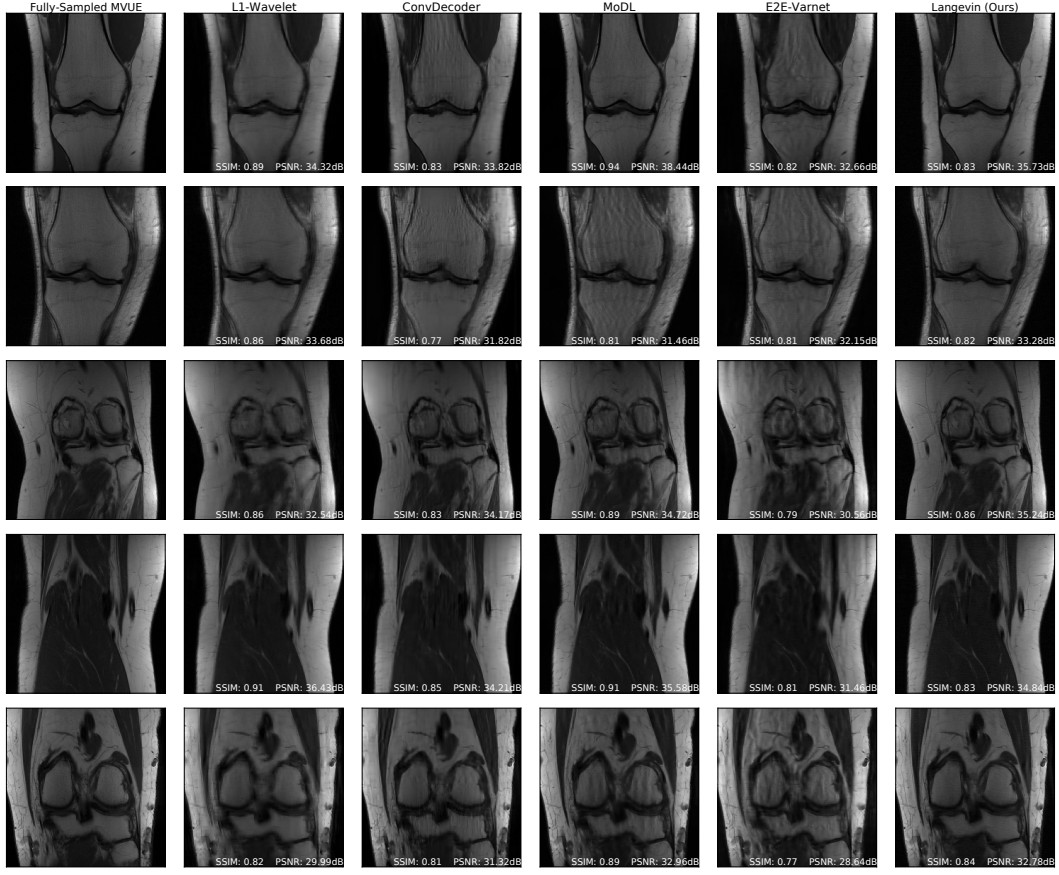

Figure 15: fastMRI knee reconstructions at an acceleration factor of $R = 4$ and a random vertical mask in k-space. All methods were trained on fastMRI brains, and this shows that our method is more robust than other methods with respect to anatomy shift.

# D Appendix: fastMRI Knee

Figure 15 and Figure 16 show further examples of proton density knee reconstructions.

Figure 18 and Figure 19 show comparisons of our method and baselines on knees with meniscus tears. Figure 17 shows uncertainty estimates from our algorithm on a knee with a meniscus tear.

Figure 20 shows PSNR and SSIM on fat-suppressed(FS) knees. Our approach is not optimal numerically, likely due to a much lower signal-to-noise ratio in FS knees than the brain training data. However, Figures 18, 19, 21, 22 show that our qualitative reconstructions are competitive, and recovers fine details (like meniscus tears) better than the deep learning baselines.

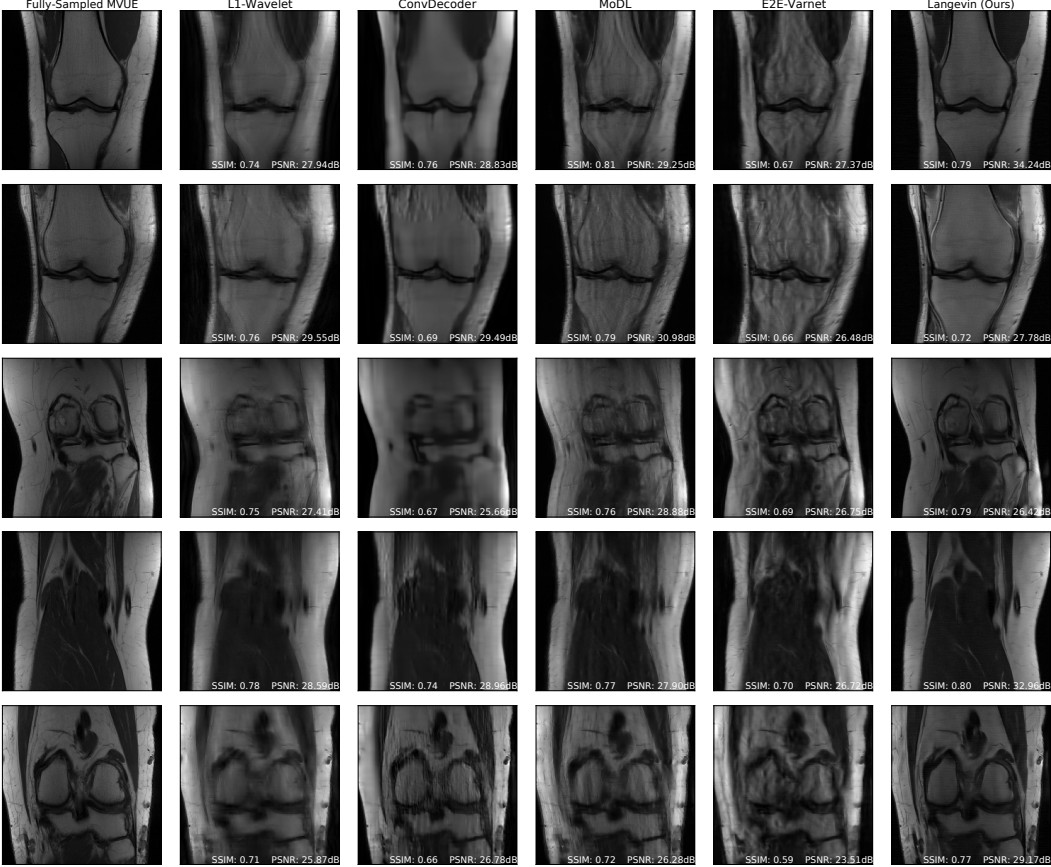

Figure 16: fastMRI knee reconstructions at an acceleration factor of $R = 8$ and a random vertical mask in k-space. All methods were trained on fastMRI brains, and this shows that our method is more robust than other methods with respect to anatomy shift.

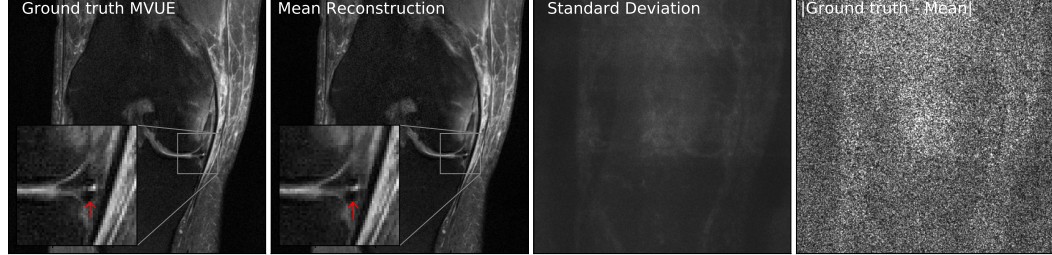

Figure 17: Our method successfully recovers fine details and can provide an estimate of the reconstruction error. The left column shows a knee from the fastMRI dataset, along with an annotated meniscus tear (indicated by red arrow in zoomed inset). Given measurements at an acceleration factor of $R = 4$, we obtain $48$ independent reconstructions via posterior sampling. The second column shows the pixel-wise average of reconstructions, the third column shows the pixel-wise standard deviation, and the fourth column shows the magnitude of the error between the ground truth and the mean reconstruction. Note that our generative prior has never seen such pathology, as it was trained on T2-weighted brain scans.

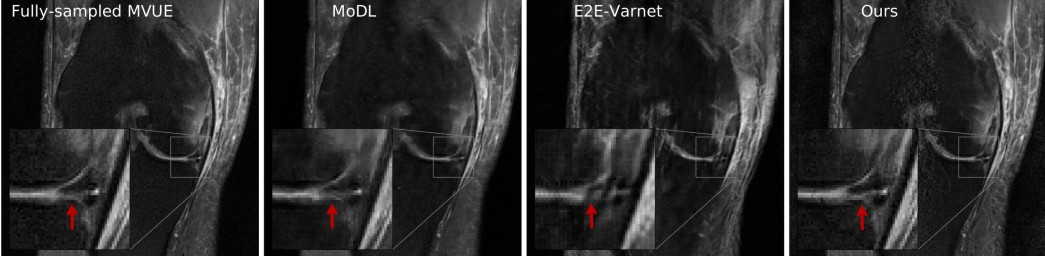

Figure 18: The left column shows a knee from the fastMRI dataset, along with an annotated meniscus tear (indicated by red arrow in zoomed inset). Given measurements at an acceleration factor of $R = 4$, we observe that our method preserves fine details better than the baselines. None of the methods have seen such a pathology, as they were all trained on T2-weighted brain scans.

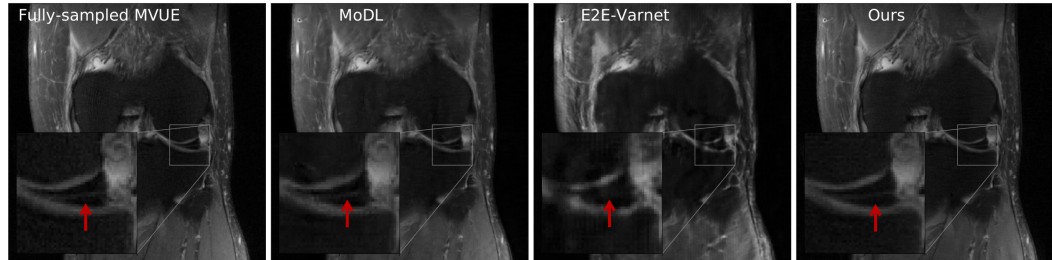

Figure 19: The left column shows a knee from the fastMRI dataset, along with an annotated meniscus tear (indicated by red arrow in zoomed inset). Given measurements at an acceleration factor of $R = 4$, we observe that our method preserves fine details better than the baselines. None of the methods have seen such a pathology, as they were all trained on T2-weighted brain scans.

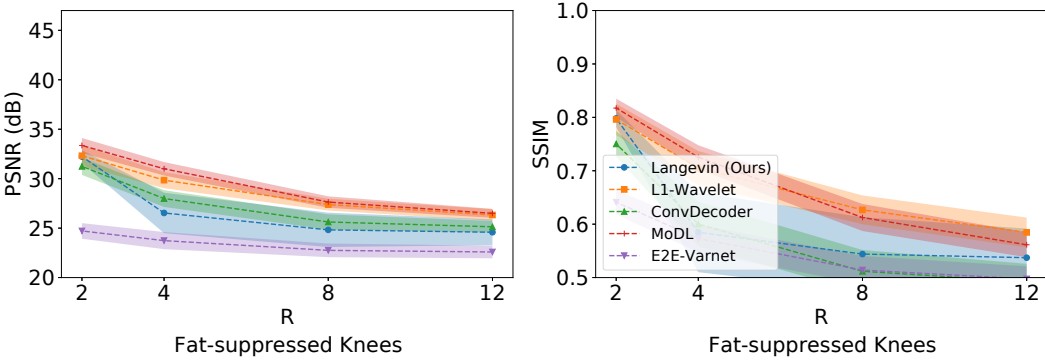

Figure 20: Average test PSNR and SSIM on fat-suppressed (FS) knees, across a range of acceleration factors $R$ and a random vertical mask in k-space. Higher $R$ indicates a smaller number of acquired measurements. All methods were trained on fastMRI brains. Our approach is not optimal numerically, likely due to a much lower signal-to-noise ratio in FS knees than the brain training data. However, Figures 18, 19, 21, 22 show that our qualitative reconstructions are competitive, and recover fine details like meniscus tears better than the deep learning baselines. Shaded regions indicate 95% confidence intervals. Note that we trained baselines on MVUE images and hence these numerical values should not be compared with those in literature trained on RSS images (see Appendix A.1 for a more detailed discussion).

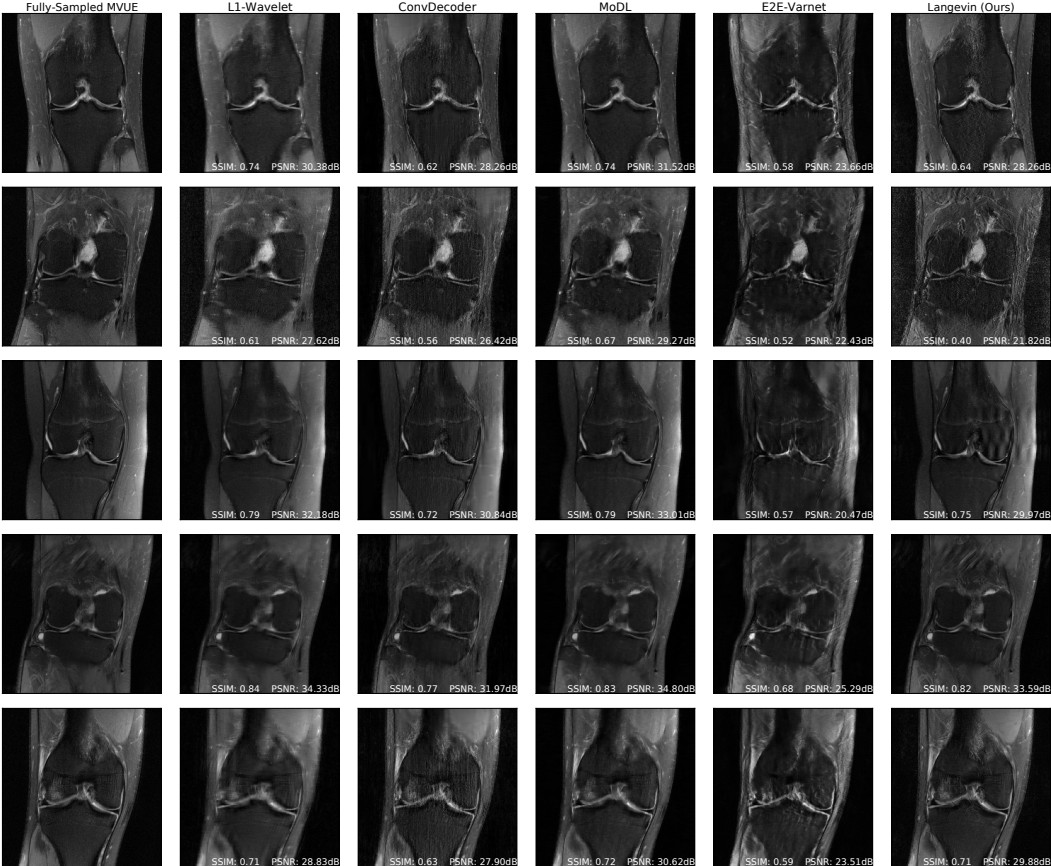

Figure 21: fastMRI fat-suppressed(FS) knee reconstructions at an acceleration factor of $R = 4$ and a random vertical mask in k-space. All methods were trained on fastMRI brains. Our approach is not optimal numerically, likely due to a much lower signal-to-noise ratio in FS knees than the brain training data. However, the reconstructions in this figure and Figures 18, 19, 22 show that our qualitative reconstructions are competitive, and recovers fine details like meniscus tears better than the deep learning baselines.

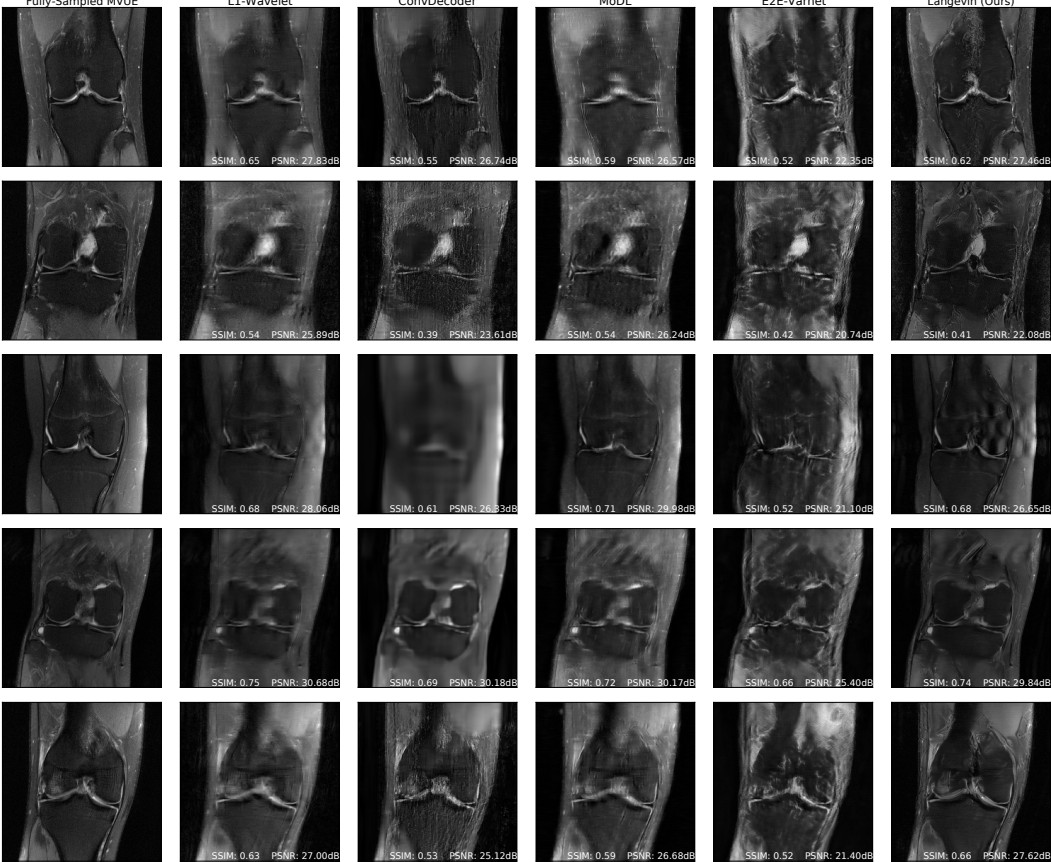

Figure 22: fastMRI fat-suppressed knee reconstructions at an acceleration factor of $R = 8$ and a random vertical mask in k-space. All methods were trained on fastMRI brains. Our approach is not optimal numerically, likely due to a much lower signal-to-noise ratio in FS knees than the brain training data. However, the reconstructions in this figure and Figures 18, 19, 21 show that our qualitative reconstructions are competitive, and recovers fine details like meniscus tears better than the deep learning baselines.

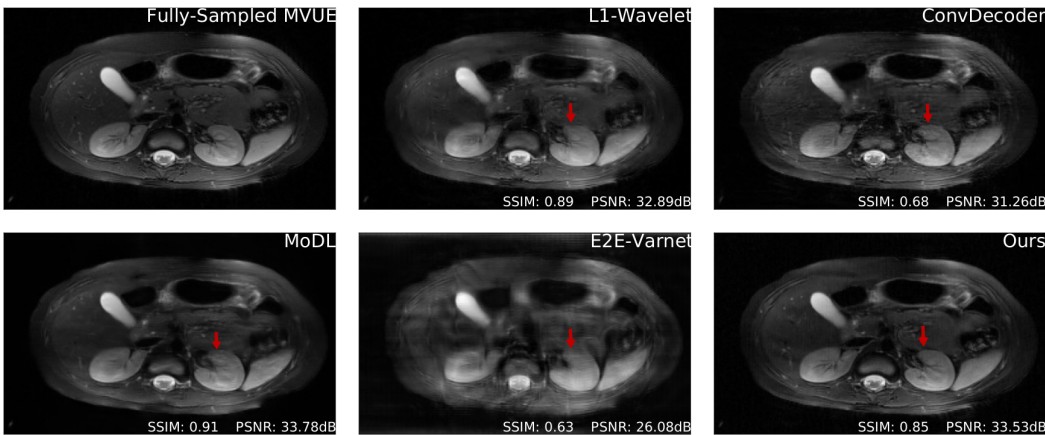

Figure 23: Comparative reconstructions of a 2D abdominal scan with uniform random under-sampling in the horizontal direction at $R = 4$. None of the methods were trained to reconstruct abdomen MRI. Our method uses a score-based generative model trained on brain images (as explained) and obtains good reconstructions. The red arrows indicate missing details or artifacts in the kidney structure.

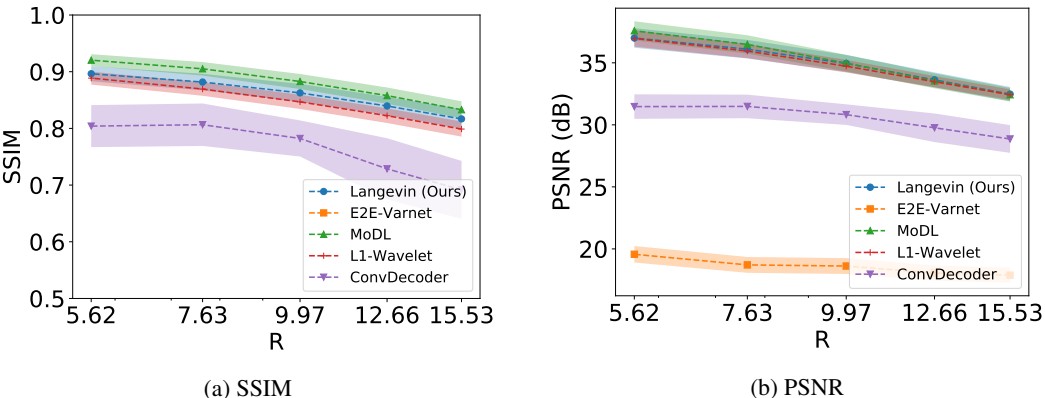

(a) SSIM                  (b) PSNR

Figure 24: Reconstruction SSIM and PSNR on Stanford Knees as a function of the acceleration $R$. This dataset is considerably different from the others, as they are 3D scans. We sample k-space measurements according to Poisson masks, which gives improved incoherence, and hence we find no statistical difference between L1-Wavelet, MoDL, and our method. Note that all hyper-parameter selection and model training was done on brains from the fastMRI dataset.

# E   Appendix: Abdomen

Figure 23 shows an additional example of a reconstructed abdominal scan. This is obtained from the same volume as the figure in the main text, and has a resolution of $158 \times 320$ voxels, but a much larger field of view, leading to a resolution shift for all models.

# F   Appendix: Stanford Knee

Figures 24 and 25 show quantitative and qualitative reconstruction under an anatomy shift induced by testing axial knee scans. In this case, we first obtain a complete three-dimensional fast spin echo (3D-FSE) knee scan from the publicly available repository at `mridata.org`. To obtain two-dimensional slices, we apply an IFFT operator on the readout axis and select $24$ equally spaced slices for evaluation. Each slice has a resolution of $320 \times 256$ pixels.

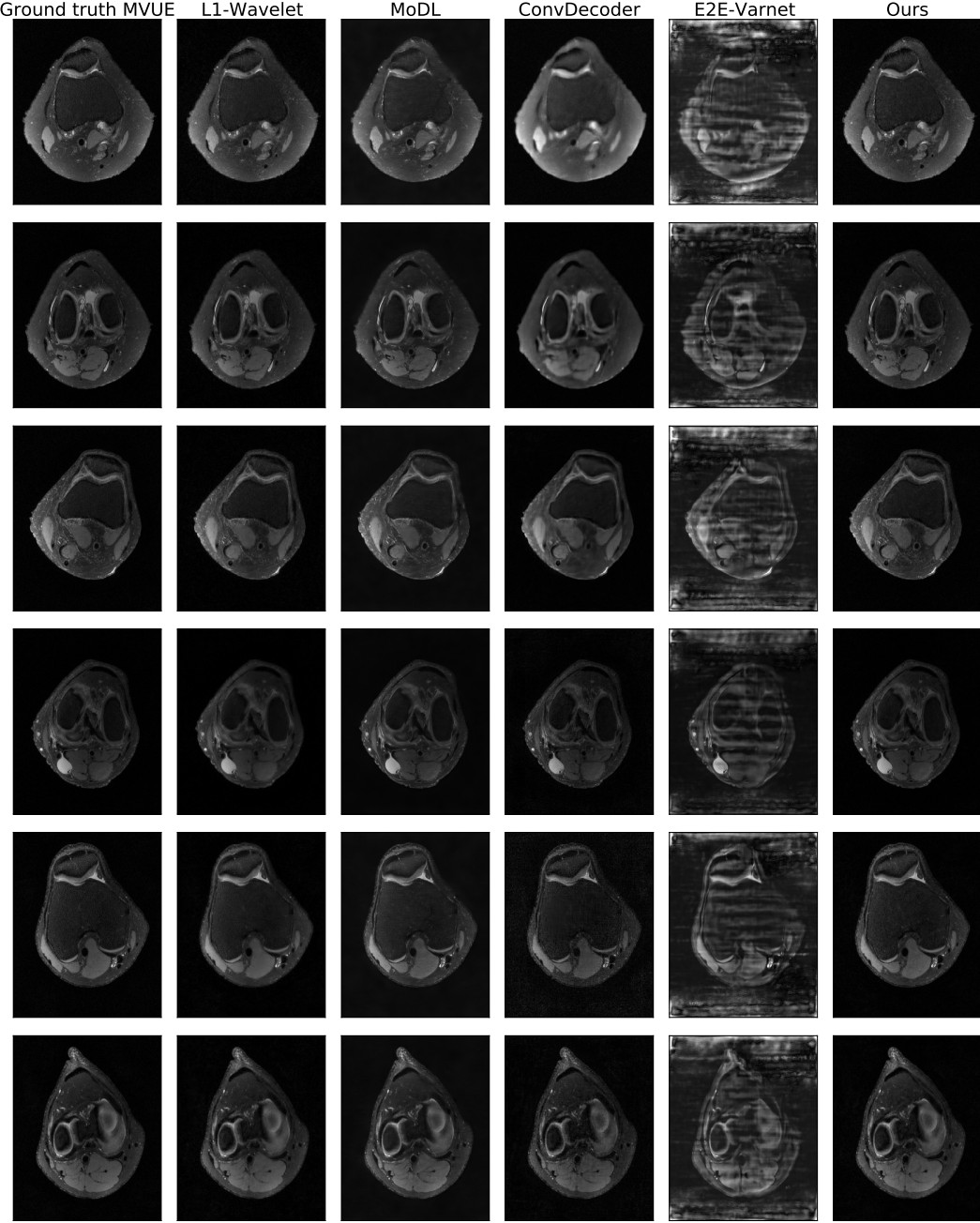

Figure 25: Qualitative reconstructions obtained by all methods on the Stanford Knees dataset at an acceleration of $R = 5.62$. This dataset is considerably different from the others, as they are 3D scans. We sample k-space measurements according to Poisson masks, which gives improved incoherence, and hence we find no statistical difference between L1-Wavelet, MoDL, and our method. Note that all hyper-parameter selection and model training was done on brains from the fastMRI dataset.

# G  Appendix: Implementation

## G.1  Score-Based Generative Model

**Training the model**  We use the implementation from `https://github.com/ermongroup/ncsnv2`. As raw MRI scans are complex valued, we changed the generator such that the output and input have two channels, one each for the real and imaginary components. We did not change the architecture otherwise.

We used the FlickrFaces (FFHQ) configs file from the NCSNv2 repo, except we set `sigma_begin` = 232, and `sigma_end` = 0.0066. This is because of the smaller number of channels in MRI when compared to FFHQ.

**Dynamic range of the data.**  MRI data exhibits a lot of variation in the dynamic range. For example, the fastMRI dataset has max pixel value on the order of $10^{-4}$, while the abdomen and Stanford knee data has max pixels on the order of $10^5$. In order to deal with this variation, during *training*, we normalize each image by the 99 percentile pixel value. During inference time, when we do not have access to the ground-truth image, we normalize the reconstruction using the 99 percentile pixel value of the *pseudo-inverse* complex image. We observe that this heuristic is sufficient to get good results.

**Invariance to image shapes.**  Due to the convolutional nature of NCSNv2, although we trained on $384 \times 384$ images, we can still apply them to knees, T1-weighted & FLAIR brains, and abdomens, although all of these have different dimension shapes.

**Hyperparameters**  We tuned our hyperparameters on two validation brain scans, at an acceleration of $R = 4$. We then reused these hyperparameters on *all anatomies, all accelerations*. Please see our GitHub link: `https://github.com/utcsilab/csgm-mri-langevin` for the hyperparameter values.

## G.2  E2E-VarNet Baseline

We use the architecture publicly available in the fastMRI official repository. The backbone for the image reconstruction network is a U-Net with a depth of four stages, and $18$ hidden channels in the first stage, for a total of $29$ million learnable parameters. This model also include a smaller deep neural network that is used to estimate the sensitivity maps. This is also a U-Net, with four stages, but only eight hidden channels after the first stage, for an additional $0.7$ million parameters. The model is trained for a number of $12$ unrolls, and separate image networks are used at each unroll.

We train this model from scratch for a number of $40$ epochs, using an Adam optimizer with default PyTorch parameters and a learning rate of $2\mathrm{e}{-4}$, decayed by $0.5$ after 20 epochs, as well as gradient clipping to a maximum magnitude of $1$. We use the fully-sampled MVUE reconstructions from the brain T2 contrast in fastMRI to train all methods. We use a batch size of $1$ and a supervised SSIM loss between the absolute values of ground truth MVUE and the absolute value of the complex output of the network at acceleration factors $R = \{3, 6\}$ (chosen with equal probability), using a vertical, equispaced sampling pattern, same as all other baselines.

Finally, it is worth mentioning that the network used to estimate the sensitivity maps explicitly uses the fully-sampled, vertical ACS region, as shown in Figure 8, both during training and inference. This makes testing with other mask patterns non-trivial for this baseline. To alleviate this, we always feed the image obtained from the *vertical* ACS region (for example, in the case of horizontal masks, we intentionally zero out other sampled lines that would fall in this region), to not introduce incoherent aliasing in this image.

## G.3  MoDL Baseline

We use the PyTorch MoDL implementation publicly available at `https://github.com/utcsilab/deep-jsense` and train a MoDL model that uses a backbone residual network with a depth of six layers, three equispaced residual connections (that feed hidden signals from the first three layers to the last three layers) and $64$ hidden channels, with a total of $220000$ trainable parameters. Unlike E2E-VarNet, the same backbone network is used across all unrolls, and the data consistency term is given by a Conjugate Gradient (CG) operator, truncated to six steps.

| Anatomy | MoDL | ConvDec | Ours |
|---------|------|---------|------|
| Knee | 1.87(0.34) | 2.97(0.18) | 1.17(0.45) |
| Abdomen | 1.87(0.76) | 2.17(0.93) | 1.97(0.71) |
| Brain | 2.00(0.82) | 2.07(0.77) | 1.93(0.85) |

Table 1: Ranking of algorithms by experts. A lower ranking is better: the best possible ranking is 1, and the worst 3. The values show the average and standard deviation (in parentheses) of the ranking for each anatomy, using a total of 30 data points (3 participants x 10 scans per anatomy).

We train MoDL for a number of six unrolls, leading to a total of 36 CG steps and six network applications in the unroll. We use the Adam optimizer with default PyTorch parameters and learning rate $2e-4$, as well as gradient clipping to a maximum magnitude of 1. We train for 15 epochs and decay the learning rate by $0.5$ after 8 epochs, using a batch size of 1 on exactly the same T2 brain scans as all methods and a supervised SSIM loss at $R = \{3, 6\}$ (chosen with equal probability) between the magnitude of the ground-truth MVUE image and the magnitude of the complex network output. We find that, although relatively small, the backbone network architecture is sufficient to achieve good in-distribution reconstruction, and serve as a strong baseline.

Since MoDL and all other methods (including ours) except E2E-VarNet, require external sensitivity map estimates to be provided to them, we use the ESPIRiT algorithm from the BART toolbox [86] without any eigenvalue cropping to estimate a single set of sensitivity maps, one for each coil.

## H   Appendix: Radiologist Study

We performed a preliminary image quality assessment experiment with two board-certified radiologists and a faculty member that uses neuro-imaging in their research.

The three external experts were not involved with our research and have performed the image quality assessment blindly. Each of them was presented with ten scans from the following anatomies and scan parameters: abdominal scans, knee scans and brain scans with a horizontal readout direction, leading to a total of 30 quality assessment questions. Note that all anatomies represent test-time distributional shifts in at least one aspect.

In each question, the experts were shown four images:

- The fully-sampled reference image, explicitly marked as "Reference".

- The results of three reconstruction algorithms at acceleration factor R=3: MoDL, ConvDecoder and our method. The order of the reconstructions was shuffled for each question, and the reconstructions were labeled as "1", "2" and "3".

We chose to compare with MoDL and ConvDecoder since these method had the best overall quantitative and qualitative (according to our own pre-assessment) robust performance. The participants were instructed to rank the three reconstructions from best to worst quality, while using the "Reference" image as a perceptual guideline. Table 1 shows the average and standard deviation (in parentheses) of the ranking for each anatomy, obtained using a total of 30 data points (3 participants x 10 scans per anatomy).

In Table 1, a lower ranking is better, the best possible ranking is 1, and the worst 3. We draw the following conclusions:

- Participants consistently ranked our method as best on the knee scans, which supports the distributional shift robustness claimed in the main paper, and detailed in Appendices D, E and A.

- Participants did not perceive a significant difference between all methods when applied to abdominal or brain scans with a horizontal phase encode direction. In the brain case, this supports the qualitative results shown in Appendix C, Figure 9.

- In the abdominal case, this partially correlates with Figure 2c, regarding the quantitative tie between our approach and MoDL.

| Anatomy | Ours vs. MoDL | Ours vs. ConvDec |
|---------|---------------|------------------|
| Knee | $1.53e-10$ | $2.77e-6$ |
| Abdomen | 0.610 | 0.340 |
| Brain | 0.767 | 0.550 |

Table 2: p-values from the Wilcoxson Rank Sum test to determine if the rankings of different algorithms are drawn from different populations. There is a significant difference in the case of knees, and no significant difference in the case of abdomens and brains.

| Anatomy | ICC2 | p-value | 95% CI |
|---------|------|---------|--------|
| Knee | 0.980 | 0.0004 | $[0.81, 1]$ |
| Abdomen | $-0.222$ | 0.576 | $[-0.89, 0.92]$ |
| Brain | $-0.818$ | 0.907 | $[-0.98, 0.59]$ |

Table 3: p-values and confidence intervals for differences in ranking between our method and baselines.

To quantify the statistical significance of the above results, we perform a Wilcoxson Rank Sum test to determine if the rankings of different algorithms are drawn from different populations. We evaluate if our proposed method leads to different rankings than MoDL and the ConvDecoder, and show the p-values in Table 2.

The results show a significant difference in the case of knees, while no significant difference is present for abdomen and brain. Finally, to evaluate inter-observer agreement between the three reviewers, we calculated the intra-class correlation (ICC) coefficient separately for each anatomy by aggregating the ten questions related to that anatomy and evaluating the ICC2 coefficient [2] in a pairwise manner at a 5% significance level.

The results are shown in Table 3, where we also include the p-value and the 95% confidence interval for the ICC2 estimate. This indicates that there exists a very strong consensus regarding the ranking on the knee anatomy, while for abdomen and brain this consensus is much weaker, which together with Table 2 indicates that the images were considered equivalent.

This preliminary image quality assessment gives additional evidence (in addition to the quantitative metrics of SSIM and PSNR) that our method maintains robustness to distribution shifts at test time. As our quantitative results show, other methods maintain robustness in some but not all cases. Due to time limitations, we were not able to ask the reviewers to evaluate every algorithm and every distribution shift including different levels of acceleration. We stress that this preliminary study is not a substitute for a rigorous clinical evaluation which is necessary before considering using our proposed method in a clinical setting.