# OpenReview forum: "Robust Compressed Sensing MRI with Deep Generative Priors"
_NeurIPS.cc/2021/Conference — NeurIPS 2021 Poster_

### Official Review · Reviewer_HfL5 · 2021-07-15

**Rating:** 7
**Confidence:** 4

**Summary:**

This paper considers the problem of compressed sensing (CS) using generative priors, which can also specifically be characterized on a multi-coil MRI setup. The algorithm itself is a more general version of the CS setup which typically considers Gaussian measurements.

The authors consider a posteriori sampling procedure to invert the measurements $y = Ax^* + w$ by estimating $\hat{x} \sim \mu(\cdot|y)$ (modeled via deep generative prior) assuming $x^* \sim \mu$, given $A$ (MRI setup or general), $y$ and zero-mean $\sigma^2$ variance Gaussian noise $w$ . Authors use an annealed Langevin Dynamics approach to sample from the posterior, which has been previously used in [73] for sampling from generative priors. Here the score function is modeled via a deep generative neural network.

Authors also present two theoretical results: one on the closeness of reconstruction $\hat{x}$ and $x^*$ under distribution shift of the deep generative prior and true distributions (assuming Gaussian A), and another on the quality of posteriori sampling as compared to any other kind of reconstruction procedure under aribitrary (not neccessarily Gaussian) measurements assuming no distributional shift.

Authors present several experiments on the fastMRI and Stanford MRI datasets and compare to baselines such as L1/wavelet, MoDL, E2EVarNet and ConvDecoder and show  improvement reconstructions on both in-distribution and out-of distribution samplings in terms of SSIM.

**Limitations And Societal Impact:**

Deep neural networks have been successfully used for MRIs and are of importance to society for furthering medical imaging research. I do not think there are negative societal impacts of this work since the authors present a clear ground truth based on MVUE (no regularization, full samples).

**Main Review:**

This paper considers a measurement agnostic approach to inverting compressively sensed images under deep generative prior. Specifically authors use the model NCSNv2 from [74] for this purpose, which is a score based generative model. This paper uses an annealed Langevin dynamics algorithm to sample from posterior $\nu$ and show experimentally that this approach works better in terms of SSIM to prior art on both in-distribution and out-of-distribution reconstructions at various acceleration rates (under sampling rates).

Through theorem 3.3, the authors present a theoretical result for $\hat{x}$ estimated according distribution $\nu$, assuming a distributional shift from the true distribution $\mu$ of $x^*$ under gaussian measurements $A$. The main claim of this result is that if $\mu,\nu$ are Wasserstein close by factor $\epsilon$, then the $\ell_2$ estimation error between $x^*$ and $\hat{x}$ is lower bounded with high probability.

Through theorem 3.4, the authors show that this posterior sampling procedure produces a reconstruction within an additional $\epsilon$-error  ($2\epsilon$) of any other reconstruction procedure that is bounded by $\epsilon$ with high probability (for any sampling process A).

The main contributions of this paper are theoretical results and the application of annealed langevin dynamics to CSGM via posteriori sampling (the algorithm itself is incremental in terms of derivation from prior art).

The paper is very well written highlighting the key contributions which is backed by their theorem statements and experiments. Experiments on Fast MRI are convincing as well. The paper can benefit from some minor changes as follows.

Figure 1 can benefit from presenting reconstruction SSIM or MSE.

Section 2.1. is the regime of experiments $compressive$? Since CSGM typically refers to $M<N$. Please make the relation between samples M and dimensionality N explicit.

Line 113 is $N_c \cdot L = M$?

Prior to Line 220, the exact formulation of the deep generative model isn't clear or explicit. Please introduce a paragraph discussing choice of deep generative prior.

**Time Spent Reviewing:**

6 hours

---

> ### Author Response · Authors · 2021-08-10
> **Author Response to Reviewer HfL5**
>
> We thank the reviewer for their thoughtful feedback and careful reading of the paper. Please find below our responses to your concerns.
>
> **Concern 1: "_Is the regime of experiments compressive? Since CSGM typically refers to $M < N$. Please make the relation between samples M and dimensionality N explicit._"**
>
> We agree that the multi-coil regime may not always be compressive, if the acceleration factor $R$ is relatively low (e.g., two). However, in practice, there is linear redundancy between the fully-sampled coils, due to the smoothness of the coil sensitivity maps and their placement, thus the dimensionality of the data is lower. To quantify this, we perform coil compression [1] on a T2 brain scan with $15$ coils and find that samples can be represented without significant SSIM degradation by using a number of $7$ virtual coils, leading to an effective dimensionality of $N=7HW$. This does mean that our considered problem is compressive when $R$ is at least $8$ for the brain data, since the number of measurements is reduced below $N$ in that case.
>
> **Concern 2: "_Figure 1 can benefit from presenting reconstruction SSIM or MSE_"**
>
> We agree and will annotate this figure with values in the revised version.
>
> **Concern 3: "_Line 113 is $N_c \cdot L = M$?_"**
>
> Yes, this is correct. To further clarify notation, the acceleration factor is defined as $R = \frac{N}{L}$.
>
> **Concern 4: "_Prior to Line 220, the exact formulation of the deep generative model isn't clear or explicit. Please introduce a paragraph discussing choice of deep generative prior._"**
>
> Thank you for the suggestion, we will include this in the revised version. This was excluded in our submission due to space limitation.
>
> **References:**
>
> [1] Huang F, Vijayakumar S, Li Y, Hertel S, Duensing G. A software channel compression technique for faster reconstruction with many channels. Magn Reson Imaging. 2008; 26:133–141. [PubMed: 17573223]

---

> ### Author Response · Authors · 2021-08-20
> **Follow up to author response**
>
> Thank you again for your review. As a gentle follow-up, based on your reading of our initial author response and the other reviews, do you have any additional concerns or comments we can address?
>
> Based on the reviewer comments and the response from Ethics Reviewer Kb9Y and Reviewer pPk8, we are conducting a preliminary image quality assessment with two board-certified radiologists and one faculty member who uses neuroimaging for their research. The three external experts are not involved with our study and will perform the image quality assessment blindly (i.e. without knowing which reconstruction algorithm was used for each image). We will post the results in the comments here, when it is complete. We emphasize that this a very preliminary experiment and not a replacement for a future detailed study that evaluates clinical diagnostic quality and this will be clear in the paper.

---

### Official Review · Reviewer_nt1J · 2021-07-16

**Rating:** 6
**Confidence:** 3

**Summary:**

This paper proposed the robust MR image reconstruction framework based on CSGM (Compressed Sensing with Generative Modeling). Compared to the other MR reconstruction algorithms, the proposed method shows consistently superior reconstruction results for both in-distribution (same k-space sampling mask/direction, same anatomy, and same contrast) and out-of-distribution (different k-space sampling direction/mask, different anatomy, and different contrast). Also, it gives the uncertainty estimation of its reconstruction results in pixel-wise manner by running multiple reconstruction samplers.

**Ethical Concerns:**

.

**Limitations And Societal Impact:**

The followings are the suggestions and my questions  to improve the paper for better understanding of readers:

- Flow of Algorithm might be helpful to the readers: Sample preparation steps such as coil-sensitivity extraction, different sampling masks/different anatomy images for the training/test phase for out-of-distribution, etc.

- Too little explanation of the main score model, NCSNv2 & backbone model, RefineNet.

- Actually, in clinic, they usually fixed the k-space sampling pattern at each acceleration rate (by MRI manufacturer) and they want to maximize the performance of reconstruction for that fixed sampling masks. Thus, the comparison results for the different sampling direction (Phase Encode for anterior-posterior VS. Phase Encode for lateral-later) seems less effective to support the usefulness of the robust algorithm. I think the different contrast images are better examples (T1w, T2w, FLAIR, etc.)

- The author mentions that their results are superior because the the comparative reconstructions show some aliasing artifacts or miss some details. However, some radiologist might score better on  the comparative one to examine the anatomy/lesion even it has some artifacts. The blind examination by the radiologist might be helpful as further work.

-- Figure.4 (a): The 'in-distribution' reconstruction performance of MoDL and E2E-Varnet seems too bad. e.g. less than SSIM 0.8 while the acceleration rate R is only 2? And the PSNR of MoDL and E2E-Varnet results are less than 25dB @only R=2 while the proposed method 40dB (appendix Fig.19(a))?

-- Figure.4 (b) : Why some results of out-of-distribution ((b)Mask Shift Brain: @R=8, R=12 ) show superior than (a) in-distribution method.

-- Figure.4 (d): Is there any discussion why the proposed method shows worse results @ higher acceleration rates, R=8 & R=12?

Minor concerns
- Please, make sure that figures&tables stand alone, complete and informative in itself.(e.g.Fig.1 what 'mask shift'/'anatomy shift' means)

- Mention about 'inverse crime' (line#214): the authors carefully slice the 3D knee scans in proper dimension to simulate the appropriate real MR acquisition. However, most of the MR image in the dataset(I think all of them), oversampled in k-space Read-Out direction(usually x2) and remove the out of the FOV part in the reconstructed images. Thus, I was wondering if the author did the proper inversion of this process (1.attach the blank bins on the image in Read-Out direction, then  Fourier Transform + retrospective sampling).

- Use capital letter correctly in references(mri, Modl, mr, etc)

- Please, specify the parameters for SSIM metric in appendix.

- Figure.3 the figures have same intensity scale? GT, mean reconstruction VS. std, error? The intensity of error images seems higher than the GT. Please, specify them.

**Main Review:**

Originality
- The most of the backbone algorithms are comes from the other works: NCSNv2, RefineNet, etc. However, the originality of this paper sufficiently comes from the how to build the robust reconstruction performance on out-of-distribution samples & its theoretical and mathematical background support.

Quality & Clarity
- The author tackles the out-of-distribution problem for MR reconstruction. And their CSGM based method makes sense for its robustness and it shows the superior MR reconstruction results for various out-of-distribution cases which are fit for their assumption and mathematical background.

Significance
- Many of MR reconstruction algorithms are still focusing on the end-to-end supervised learning (accelerated/sampled data to full-sampled data) and it results to the poor performance on out-of distribution datasets while it results to the excellent performance on in-distribution datasets. Since the proposed method shows impressive robustness of various out-of-distribution datasets (sampling direction/anatomy/contrast), it might give positive impact on real clinical MR acceleration fields.

**Time Spent Reviewing:**

11

---

> ### Author Response · Authors · 2021-08-10
> **Author Response to Reviewer nt1J**
>
> We thank the reviewer for their thoughtful feedback and careful reading of the paper. Please find below our responses to your concerns.
>
> **Concern 1: "_Flow of Algorithm might be helpful to the readers..._"**
>
> We did not include this in the paper due to space limitations, however the code does specify all the pre-processing. We shall reorganize the paper to include these details.
>
> **Concern 2: "_Too little explanation of the main score model, NCSNv2 \& backbone model, RefineNet._"**
>
> This was also excluded due to space limitations, we shall include it in future versions.
>
> **Concern 3: "_Actually, in clinic, they usually fixed the k-space sampling pattern at each acceleration rate (by MRI manufacturer) and they want to maximize the performance of reconstruction for that fixed sampling masks. Thus, the comparison results for the different sampling direction (Phase Encode for anterior-posterior VS. Phase Encode for lateral-later) seems less effective to support the usefulness of the robust algorithm. I think the different contrast images are better examples (T1w, T2w, FLAIR, etc.)_"**
>
> While this may be true for the particular example of phase encode direction, we argue that robustness is very valuable from a clinical perspective.
> In many clinical situations, the acceleration may change due to different slice prescriptions (e.g., different field of view because of the patient body size, while keeping the same spatial resolution) and scan time requirements.
> In other applications such as interactive MRI [1-3], a clinician may vary the acceleration after an initial acquisition or during a complicated procedure, such as cardiac surgery.
>
> This can lead to a large set of possible k-space sampling patterns across different anatomies. While an ensemble of end-to-end models could be trained to accommodate all possible patterns, we are not aware of prior work that suggests this is practical, since it is unclear at what granularity models should be trained. We will incorporate this justification in our introduction section.
>
> Additionally, we point out that Appendix B.2 already contains results for evaluating all methods on different brain contrasts (T1, T2, and FLAIR), and shows our method is robust to this type of shift as well. We will point this out in the main text.
>
> **Concern 4: "_The author mentions that their results are superior ... However, some radiologist might score better on the comparative one to examine the anatomy/lesion even it has some artifacts. The blind examination by the radiologist might be helpful as further work._"**
>
> We thank the reviewer for pointing this out, and we agree that diagnostic evaluation by a clinician is an important next step following this proof-of-principle feasibility study. We will emphasize this point in the revision of our paper.
>
> **Concern 5: "_Figure.4 (a), Fig 19(a): The 'in-distribution' reconstruction performance of MoDL and E2E-Varnet seems too bad..._"**
>
> We restate here the answer given to Reviewer zR76 Concern 1 & 2, since it concerns the same aspect of performance metrics.
>
> As Reviewer zR76 has pointed out, the difference in ranking comes from training and evaluating on MVUE instead of RSS images. This is a design choice that we have made, since our goal is to compare with a wide range of previous methods in a fair way.
>
> Algorithms that output a complex-valued image (such as ours and L1-Wavelet) as a solution to the optimization in Eqn (2) will artificially perform worse (w.r.t. E2E methods) when compared to the RSS ground truth, even when the output is of similar or higher quality, due to the bias in the RSS.
>
> Since there is no way to obtain a good RSS score with these algorithms, this justifies our choice to train and evaluate all methods on MVUE (or, in the sole case of E2E-VarNet, fine-tuning the pre-trained model due to computational constraints). This also explains the difference in ranking, and why MoDL performs better than previously reported, since we were able to train this baseline from scratch on MVUE images.
>
> To the best of our knowledge, a rigorous comparison between end-to-end models trained on RSS or MVUE images has not been made in prior work. The recent work of [KH] has also discussed this point. We will add further clarifications on this in the revised paper. To illustrate our claim of incompatibility between the two estimates, as well as the importance of qualitative inspection, we provide two simple, easy-to-verify examples, which will be included in the revised supplementary material:
> - We compare the fully sampled MVUE reconstruction (with ESPiRIT estimated maps) with the fully sampled RSS reconstruction, on T2 brain scans: we find that the SSIM is slightly larger than 0.8. This is a large penalty, even though the two images are virtually indistinguishable and known to be clinically equivalent (see discussions of SENSE vs. GRAPPA in [KH]). This would unfairly penalize the family of methods that explicitly solve the inverse problem. Since E2E methods can be trained to target the MVUE directly, this justifies our choice for using the MVUE as the reference image.
> - We point to the public knee fastMRI leaderboard at https://fastmri.org/leaderboards. Selecting "Multi-coil Knee" and "4x" acceleration, we inspect the two following submissions:
>     1. "zero-filling", which does zero-filling RSS reconstruction, has an SSIM of 0.804 and considerable artifacts.
>     1. "Baseline Classical Reconstruction Model", which applies compressed sensing with the ESPiRIT algorithm, has a much poorer SSIM score of 0.6275, but produces qualitatively superior reconstructions.
>
> [KH] Hammernik, K, Schlemper, J, Qin, C, et al. Systematic evaluation of iterative deep neural networks for fast parallel MRI reconstruction with sensitivity-weighted coil combination. Magn Reson Med. 2021; 86: 1859– 1872. https://doi.org/10.1002/mrm.28827
>
> **Concern 6: "_Figure.4 (b) : Why some results of out-of-distribution ((b)Mask Shift Brain: @R=8, R=12 ) show superior than (a) in-distribution method._"**
>
> Note that the algorithms that show an improvement (ours, L1-Wavelet and ConvDecoder) do not depend on knowledge of the sampling masks. This may indicate that horizontal masks are more favorable at very high acceleration factors for the considered anatomies.
>
> **Concern 7: "_Figure.4 (d): Is there any discussion why the proposed method shows worse results @ higher acceleration rates, R=8 \& R=12?_"**
>
> We will include this in the revised version. On fastMRI knees (the dataset in Fig 4d), all the methods perform poorly at very high accelerations, and our method degrades to the same level as the baselines.
>
> **Concern 8: Regarding inverse crime and proper inversion on 3D knee scans**
>
> We thank the reviewer for double-checking this, and confirm that we have performed the slicing correctly. The main difference here comes from the fact that the data from [4] are acquired with a General Electric (GE) MRI scanner, which does **not** by default perform oversampling by a factor two in the readout direction. Note this is also the case for the abdomen scans.
>
> **Other minor concerns** We thank the reviewer for the suggestions, we will incorporate this feedback in future versions.
>
> **References:**
>
> [1] Kerr, A. B. et al. "Real-time Interactive MRI on a Conventional Scanner", Magnetic Resonance in Medicine, Vol. 38, Issue 3, 1997. https://onlinelibrary.wiley.com/doi/10.1002/mrm.1910380303
>
> [2] Sumbul, Uygar, Juan M. Santos, and John M. Pauly. "Improved time series reconstruction for dynamic magnetic resonance imaging." IEEE transactions on medical imaging 28.7 (2009.
>
> [3] Campbell‐Washburn, Adrienne E., et al. "Real‐time MRI guidance of cardiac interventions." Journal of Magnetic Resonance Imaging 46.4 (2017): 935-950.
>
> [4] http://mridata.org/list?project=Stanford\%20Fullysampled\%203D\%20FSE\%20Knees

---

> ### Author Response · Authors · 2021-08-20
> **Follow up to author response**
>
> Thank you again for your review. As a gentle follow-up, based on your reading of our initial author response and the other reviews, do you have any additional concerns or comments we can address?
>
> Based on the reviewer comments and the response from Ethics Reviewer Kb9Y and Reviewer pPk8, we are conducting a preliminary image quality assessment with two board-certified radiologists and one faculty member who uses neuroimaging for their research. The three external experts are not involved with our study and will perform the image quality assessment blindly (i.e. without knowing which reconstruction algorithm was used for each image). We will post the results in the comments here, when it is complete. We emphasize that this a very preliminary experiment and not a replacement for a future detailed study that evaluates clinical diagnostic quality and this will be clear in the paper.

---

> ### Author Response · Authors · 2021-08-26
> **Follow-up to Reviewer nt1J regarding Expert Quality Assessment**
>
> We thank Reviewer nt1J once again for their valuable suggestions and would like to kindly point out that we have submitted the results of a preliminary image quality assessment experiment as a top-level comment.
>
> We have conducted this assessment with two board-certified radiologists and a faculty member that are not involved in our research, and have found that our algorithm is ranked as best for knee scans, and ties with the baselines for abdominal and brain scans, supporting our robustness claims in the paper. For more details, please see our high-level comment and we welcome any feedback.

---

> ### Comment · Reviewer_nt1J · 2021-09-02
> **Follow up after author response**
>
> Thank you for the responses provided to me and other reviewers. Some answers have helped me better understand the paper.
>
> Concern 1-2
> I understand the limitation of the space.
> Concern 3
> My suggestion was emphasizing the results of the different contrast images in the Appendix. Thanks for the response and the clarifications.
> Concern 4
> I checked the results that you post, ‘Expert Quality Assessment’. That could be the great supportive evidence to prove the superiority of the proposed algorithm for the reviewers in clinics. I really appreciate your efforts.
> Concern 5
> I also agreed that evaluating on MVUE images makes a lot of sense for the comparison with a wide range of previous methods. However, the other methods haven’t been tried by that way (MVUE) for the best results. Adding some rows of the original results with ‘*’ which the original paper reports, and explain that your comparison method might be might be the one way to state fair way.
> Concern 8.
> Thank you for your response and the clarifications.
>
> I don't have any more concerns and thanks for your responses. I would recommend the paper for acceptance.

---

### Official Review · Reviewer_zR76 · 2021-07-16

**Rating:** 6
**Confidence:** 4

**Summary:**

The paper applies deep generative priors successfully to accelerated MRI on clinical data.

The paper trains a score-based generative prior on brain scans from the fastMRI dataset. To recover an image from sub-sampled multi-coil data, the paper imposes the prior by performing posterior sampling via Langevian dynamics. The paper provides numerical results showing that this approach gives very good image quality. Besides the standard setup of in-distribution evaluation, the paper also provides numerical results for different sampling patterns and for a distribution shift from one anatomy to another, and again shows very good results even when applied to one anatomy and trained on another anatomy.
The paper also emphasizes that their method can obtain multiple reconstructions with different random initializations which can be used to quantify the uncertainty of the reconstructions.
Finally, the paper provides theoretical results for posterior sampling.

**Ethical Concerns:**

None.

**Limitations And Societal Impact:**

Yes.

**Main Review:**

Comments:
1. The ranking of algorithms on in-distribution images (Section 4.1, Figure 4(a)) is inconsistent with the results in the literature. In the FastMRI leaderboard, Varnet outperforms ConvDecoder (Deep Decoder) on the Brain dataset, while in Figure 4a, Varnet is by 0.1 in SSIM  worse than ConvDecoder. This is a significant difference.
Perhaps the reason is that the paper evaluates on MVUE images, but uses the VarNet pre-trained on RSS images (but fine-tuned on MVUE images).
In any case, it is important to understand where this difference comes from, because if in fact, the Varnet outperforms ConvDecoder as consitently reported in the literature, then Varnet would outperform Lagevian (i.e., the method in the paper), since ConvDecoder is very close to Lagevian. This in turn would change a major conclusion of the paper saying that 'we achieve competitive performance compared to end-to-end deep learning methods when the test-time data are sampled within the distribution'.

2. Overall the experimental setup is very good and well described. However, why use MVUE images instead of RSS as the ground truth? The paper's numerical results are based on the FastMRI dataset and the FastMRI competition and all associated results are based on using the RSS images as groundtruth. Choosing MVUE as the groundtruth makes a comparison to the literature difficult and also leads to the complication that the VarNet used in the paper under review is trained on the RSS images, but then applied to a MVUE images (albeit it is fine-tuned on MVUE images).

3. In the distribution shift section: It would be interesting to specify whether the results are obtained by applying the CSGM method with the same hyperparameters for changing sampling patterns, or if the paper tunes the hyperparameters of the methods on the new sampling patterns. This is important to know, as [20] has shown that hyperparameter tuning on the target distribution significantly impacts the performance on out-of-distribution data.

4. The paper emphasizes that the Lagevian sampling method can obtain multiple reconstructions with different random initializations which can be used to quantify the uncertainty of the reconstructions. The paper writes in the corresponding Section 4.3 'as shown in Fig. 3, the pixel-wise standard deviation is a good estimate of the ground truth error': I really don't see this conclusion from Figure 3. It would be great to have a quantitative result for this statement.

5. Theorem 3.4 states that if there is an algorithm that reconstructs $x^\ast$ drawn from some distribution $\mu$ from $y = A(x^\ast)$ with $\epsilon$ accuracy, then posterior sampling succeeds with $2\epsilon$-accuracy. That's an interesting statement, but it would be good to also discuss the catch: for a given forward operator, it is typically difficult to check whether an $\epsilon$-accurate algorithm exists, and we don't have access to the distribution of $\mu$, but only to an estimate through the generative model. Also, it would be great if the paper could comment on whether the Langevin Dynamics converges to a sample from the posterior distribution, in order to clarify whether the recovery results Thm 3.3 and 3.4 apply to the algorithm that is used in the numerical results section.

Significance and originality:
The paper provides evidence that imposing a prior (i.e., recovery via the SCGM framework) can give very competitive results for clinical MRI. This is very interesting, as prior works did not achieve competitive results for real-world medical imaging data by imposing a generative prior. Imposing a prior has a number of advantages, in particular, that the method requires no retraining when changing the forward model. If the SCGM framework can really give results that are competitive to the state-of-the-art end-to-end neural networks, this would be a significant development.

Quality and clarity:
The paper is well written overall and describes the setup and experiments well along with the assumption and the setup (apart from a few details pointed out above and in the minor comments below). The paper evaluates the method well, and the experimental setup makes a lot of sense, the only major concern outlined in comment 1 above is the significant discrepancy of the relative ranking of algorithms relative to the literature.

Summary:
I really enjoyed the paper and would be happy to see a (revised version) at the conference. My reason for not rating it higher, my concern in explained in comment 1. If this can be convincingly resolved, without the need to change the main claim of the paper, the paper would be a clear accept in my opinion, and I'll change my rating accordingly then.



Minor comments:
- Fig 1: Would be good to state the acceleration factor in the caption for reference, and whether this is single or multi-coil reconstruction.
- The paper considers 'test time sampling patterns shifts': I fully agree that a major advantage of the CSGM framework is that it can be easily adapted to changes in the sample process, without the need of re-training. I wouldn't call this a distribution shift as the image distribution doesn't change, but again this is a minor point.
- A related work is 'Compressible Latent-Space Invertible Networks for Generative Model-Constrained Image Reconstruction' which also showed good results on the FastMRI dataset, albeit not on the original k-space data and not on multi-coil data, and with a different algorithm, but also by imposing a generative prior via the CSGM framework.

**Time Spent Reviewing:**

7

---

> ### Author Response · Authors · 2021-08-10
> **Author Response to Reviewer zR76**
>
> We thank the reviewer for their thoughtful feedback and careful reading of the paper. Please find below our responses to your concerns.
>
> **Concern 1 & 2: _Discussion of MVUE vs. RSS and Algorithm Ranking_**
>
> As the reviewer has pointed out, the difference in ranking comes from training and evaluating on MVUE instead of RSS images. This is a design choice that we have made, since our goal is to compare with a wide range of previous methods in a fair way.
>
> Algorithms that output a complex-valued image (such as ours and L1-Wavelet) as a solution to the optimization in Eqn (2) will artificially perform worse (w.r.t. E2E methods) when compared to the RSS ground truth, even when the output is of similar or higher quality, due to the bias in the RSS.
>
> Since there is no way to obtain a good RSS score with these algorithms, this justifies our choice to train and evaluate all methods on MVUE (or, in the sole case of E2E-VarNet, fine-tuning the pre-trained model due to computational constraints). This also explains the difference in ranking, and why MoDL performs better than previously reported, since we were able to train this baseline from scratch on MVUE images.
>
> To the best of our knowledge, a rigorous comparison between end-to-end models trained on RSS or MVUE images has not been made in prior work. The recent work of [KH] has also discussed this point. We will add further clarifications on this in the revised paper. To illustrate our claim of incompatibility between the two estimates, as well as the importance of qualitative inspection, we provide two simple, easy-to-verify examples, which will be included in the revised supplementary material:
> - We compare the fully sampled MVUE reconstruction (with ESPiRIT estimated maps) with the fully sampled RSS reconstruction, on T2 brain scans: we find that the SSIM is slightly larger than 0.8. This is a large penalty, even though the two images are virtually indistinguishable and known to be clinically equivalent (see discussions of SENSE vs. GRAPPA in [KH]). This would unfairly penalize the family of methods that explicitly solve the inverse problem. Since E2E methods can be trained to target the MVUE directly, this justifies our choice for using the MVUE as the reference image.
> - We point to the public knee fastMRI leaderboard at https://fastmri.org/leaderboards. Selecting "Multi-coil Knee" and "4x" acceleration, we inspect the two following submissions:
>     1. "zero-filling", which does zero-filling RSS reconstruction, has an SSIM of 0.804 and considerable artifacts.
>     1. "Baseline Classical Reconstruction Model", which applies compressed sensing with the ESPiRIT algorithm, has a much poorer SSIM score of 0.6275, but produces qualitatively superior reconstructions.
>
> [KH] Hammernik, K, Schlemper, J, Qin, C, et al. Systematic evaluation of iterative deep neural networks for fast parallel MRI reconstruction with sensitivity-weighted coil combination. Magn Reson Med. 2021; 86: 1859– 1872. https://doi.org/10.1002/mrm.28827
>
> **Concern 3: "_In the distribution shift section: It would be interesting to specify whether the results are obtained by applying the CSGM method with the same hyperparameters for changing sampling patterns, or if the paper tunes the hyperparameters of the methods on the new sampling patterns. This is important to know, as [20] has shown that hyperparameter tuning on the target distribution significantly impacts the performance on out-of-distribution data._"**
>
> We do not tune hyperparameters under any mask/anatomy shift -- the same hyperparameters are used in- and out-of-distribution, and they are found using only in-distribution validation data. This is mentioned in the paper in Section 2.2 and the end of Section F.2 in the Appendix. We will make this more explicit in the main paper.
>
> **Concern 4: "_...The paper writes in the corresponding Section 4.3 'as shown in Fig. 3, the pixel-wise standard deviation is a good estimate of the ground truth error': I really don't see this conclusion from Figure 3. It would be great to have a quantitative result for this statement._"**
>
> Thank you for the suggestion, we will repeat the experiment on more samples and show concordance correlation coefficients (CCC) between the estimated error and ground-truth error.
>
> **Concern 5, Part 1: "_Theorem 3.4 states ... That's an interesting statement, but it would be good to also discuss the catch: for a given forward operator, it is typically difficult to check whether an $\epsilon$-accurate algorithm exists, and we don't have access to the distribution of $\mu$, but only to an estimate through the generative model._"**
>
> We agree, and shall include a discussion about this in the theory section. The current remark under Theorem 3.4 discusses the caveat of merging Theorem 3.3 and Theorem 3.4. We will include the caveats mentioned by the reviewer, and mention examples of forward operators and prior distributions where we know $\epsilon$-accurate recovery is possible even under distribution mismatch.
>
> **Concern 5, Part 2: "_...comment on whether the Langevin Dynamics converges to a sample from the posterior distribution, in order to clarify whether the recovery results Thm 3.3 and 3.4 apply to the algorithm that is used in the numerical results section._"**
>
> It is known [4] that Langevin dynamics will asymptotically converge to a sample from the posterior distribution, i.e., if the step size tends to $0$ and the number of steps goes to $\infty$, then the sample will be from the posterior distribution.
>
> Non-asymptotic versions of this result is an ongoing area of research, and theoretical guarantees exist for certain special classes of distributions (see Part 1, 2, 3 of this excellent blog post [5]). For the distributions considered in our paper, there are no known theoretical guarantees, but there is growing empirical evidence that annealed Langevin dynamics is a good approximation to Posterior Sampling. Synthetic experiments in [1] show examples on a simple two-dimensional distribution where annealed Langevin dynamics using a score-based model (trained on the ground truth distribution) produces empirical samples closely resembling the true distribution. For high-dimensional distributions and inverse problems, experiments in [2] and [3] show that annealed Langevin dynamics produces estimates that obey the statistics of the posterior distribution, and this holds true for score-based generative models, invertible / flow-based generative models, and GANs.
>
> Finally, we'd like to emphasize that our theoretical results can be modified to accommodate error between posterior sampling and Langevin dynamics. If Langevin dynamics produces a sample that is $\beta$-close to posterior sampling in Total Variation distance, then the probability of recovery failure increases by $\beta$.
>
> [1] Song, Yang, and Stefano Ermon. "Generative Modeling by Estimating Gradients of the Data Distribution." In Proceedings of the 33rd Annual Conference on Neural Information Processing Systems. 2019.
>
> [2] Jalal, Ajil, Sushrut Karmalkar, Alexandros G. Dimakis, and Eric Price. "Instance-Optimal Compressed Sensing via Posterior Sampling." arXiv preprint arXiv:2106.11438 (2021).
>
> [3] Jalal, Ajil, Sushrut Karmalkar, Jessica Hoffmann, Alex Dimakis, and Eric Price. "Fairness for Image Generation with Uncertain Sensitive Attributes." In International Conference on Machine Learning, pp. 4721-4732. PMLR, 2021.
>
> [4] Dominique Bakry and Michel Émery. Diffusions hypercontractives. In Seminaire de probabilités XIX 1983/84, pages 177–206. Springer, 1985.
>
> [5] https://www.offconvex.org/2021/03/12/beyondlogconcave3/
>
> **Minor comments and additional reference** Thank you for the suggestions, we shall include them in future versions of the paper.

---

> ### Author Response · Authors · 2021-08-20
> **Follow-up to author response**
>
> Thank you again for your review. As a gentle follow-up, based on your reading of our initial author response and the other reviews, do you have any additional concerns or comments we can address?
>
> Based on the reviewer comments and the response from Ethics Reviewer Kb9Y and Reviewer pPk8, we are conducting a preliminary image quality assessment with two board-certified radiologists and one faculty member who uses neuroimaging for their research. The three external experts are not involved with our study and will perform the image quality assessment blindly (i.e. without knowing which reconstruction algorithm was used for each image). We will post the results in the comments here, when it is complete. We emphasize that this a very preliminary experiment and not a replacement for a future detailed study that evaluates clinical diagnostic quality and this will be clear in the paper.

---

> ### Comment · Reviewer_zR76 · 2021-08-26
> **Response**
>
> Thanks for the response and the clarifications, I really appreciate it!
>
> I agree that evaluating on MVUE images makes a lot of sense, in particular given the method the paper under review considers.
>
> But to put the paper's method into context, we still need a good comparison to end-to-end networks: The current results (Figure 4) indicate that Langevian significantly outperforms end-to-end approaches (specifically E2E VarNet) (by more than 0.1 SSIM which is a lot!). However, it looks like much if not all of this difference in performance can likely be attributed to the fact that VarNet is trained on RSS images and only fine-tuned on a few MVUE images. Specifically, for in-distribution images, the results of E2E-varnet and Langevian both look very good, but E2E looks slightly better (see the details on the brain on the very top of the image, E2E obtains those a bit better), indicating that the numbers are off because of the way E2E is trained.
>
> It's easy to fix this, by adding a comparison where either E2E-varnet or U-net is trained on MVUE images and not just finetuned on them. E2E varnet doesn't take more than a few days to train and U-net not more than two days. Even an E2E approach is slightly better after this fix than the Lagevian approach, this wouldn't make the paper any less interesting, it's just important to have this comparison.
>
> Other than that I don't have open concerns and with this change (i.e., with adding a fair comparison) I would recommend the paper for acceptance.

---

> > ### Author Response · Authors · 2021-08-26
> > **MoDL baseline is trained on MVUE**
> >
> > We thank the reviewer for their response and for agreeing that MVUE is a suitable metric in this case.
> >
> > Regarding end-to-end models trained on MVUE from scratch, we would like to point out that such a model is already present in our paper through the MoDL [1] baseline, which is an end-to-end unrolled method that we trained from scratch for our MVUE setting. We apologize for the confusion and if this point was not clear from the paper and our original response - it is only E2E-VarNet that was pretrained on RSS and we fine-tuned, because of computational constraints.
> >
> > This also correlates with what the reviewer has suggested about the ordering of the methods, and explains why MoDL surpasses E2E-VarNet on the quantitative results.
> >
> > Please let us know if this resolves the issue of including a fair comparison with an end-to-end method. If the reviewer feels that this is still not sufficient, we could train an E2E-VarNet model from scratch, but this would likely take at least one week, due to large model size.
> >
> > We will update the paper to better clarify that MoDL is an end-to-end method trained from scratch on MVUE images for a fair comparison.
> >
> > References
> >
> > [1] Aggarwal, Hemant K., Merry P. Mani, and Mathews Jacob. "MoDL: Model-based deep learning architecture for inverse problems." IEEE transactions on medical imaging 38.2 (2018): 394-405.

---

### Official Review · Reviewer_pPk8 · 2021-07-22

**Rating:** 4
**Confidence:** 4

**Summary:**

This paper propose to use score-based generative models for robust compressed MRI. The paper claims that CSGM framework performs poorly on out-of-distribution samples, and the proposed posterior sampling-based method is robust to changes in the image distribution and measurement process.

**Ethical Concerns:**

MRI is a popular imaging scheme and we need thorough analysis and test to make claims about generalization/accuracy of the proposed methods. I did not find an adequate discussion on ethical concerns in the paper. The paper actually claim that they are the first to show results on clinical data, but they probably should rephrase it as publicly available real data.

**Ethics Review Area:**

["Responsible Research Practice (e.g., IRB, documentation, research ethics)"]

**Limitations And Societal Impact:**

Brief discussion on limitation in the conclusion section.

**Main Review:**

I list my main concerns below.

- The main contribution of this paper seems to be in using score-based generative models for MRI. The main approach/algorithm is based on prior work that uses a score-based generative model to perform Langevian dynamics for posterior sampling.
- The paper claims that they are the first to apply CSGM framework on clinical MRI data. This seems to be a strong claim because several other paper have used fastMRI data with generative model-based reconstruction.
- The paper also discussed the challenges related to shifts in the distribution of images and measurements.
- The challenge of shift in image distribution at train and test times seems real and important to me, but I do not understand why measurement distribution is of any concern/relevance. It seems obvious to me that the generative model-based methods can work with any measurement pattern. This is different from end-to-end networks that learn to map measurements to the images, and difference between measurement patterns at training and testing is a concern.
- The problem of mismatch between train and test images of generative models is important. This paper provides empirical results that show the proposed method is robust to such mismatch. At the same time, most of other methods they compare against in Figure 4c also see robust to such image distribution mismatch. I wonder what is the significance of the proposed method in that case?
- The paper does not compare with the original CSGM framework in which you use a trained generative model and solve an optimization problem similar to the one in Eq. 2.
- SSIM is a metric often used for natural image quality assessment. It is not necessarily a good metric for MR images. Please report PSNR as well.
- Theoretical results: I feel there is a mismatch between the theoretical and empirical results in the paper. Gaussian measurements do not exist in MRI, so what is the purpose of discussing those results? Theorem 3.4 talks about general operators, but it is also a generic result that does not take into account any specific component of the proposed generative model-based method.


**Needs Ethics Review:**

Yes

**Time Spent Reviewing:**

5

---

> ### Author Response · Authors · 2021-08-10
> **Author Response to Reviewer pPk8**
>
> We thank the reviewer for their thoughtful feedback and careful reading of the paper. Please find below our responses to your concerns.
>
> **Concern 2: "_The paper claims that they are the first to apply CSGM framework on clinical MRI data. This seems to be a strong claim because several other paper have used fastMRI data with generative model-based reconstruction._"**
>
> Could the reviewer please clarify what papers are being referred to?  To the best of our knowledge, we are the first to successfully apply the CSGM framework on clinical MRI data. The work in [1] uses CSGM, but uses (i) DICOM magnitude-only images, (ii) simulated measurements using a Discrete Fourier Transform, (iii) a single coil, and (iv) a prior-image as side information, and hence is not representative of clinical multi-coil MRI. Alternatively, the work in [2] uses CGSM with a WGAN model applied to fastMRI, but suffers from poor performance, and a lack of reproducibility, since key hyperparameters and stopping conditions were omitted and no code was provided.
>
> The discussion related to these papers is already included in our paper, in lines 37 and 96, respectively. Our claim is also backed by the very recent work in [3], which recognizes that "this class of neural network-based reconstruction methods has not been applied to MRI", referring to generative models.
>
> **Concern 1: "_The main contribution of this paper seems to be in using score-based generative models for MRI. The main approach/algorithm is based on prior work that uses a score-based generative model to perform Langevian dynamics for posterior sampling._"**
>
> The primary objective of this paper is to (i) demonstrate a _successful_ CSGM style algorithm for MRI, and (ii) study its robustness and performance wrt SOTA methods.
> The significance / contribution of this paper is that: (i) it shows posterior sampling retains the flexiblility of CSGM style methods, (ii) it shows that posterior sampling is robust in different settings, and (iii) algorithms that use generative priors are an important class of methods that have not been successful on MRI[3].
>
>
>
> **Concern 3: "_I do not understand why measurement distribution is of any concern/relevance. It seems obvious to me that the generative model-based methods can work with any measurement pattern..._"**
>
> Though it may be obvious to those deeply familiar with generative modelling for inverse problems, an unfamiliar reader may not find this obvious. The fact that our method does not require knowledge of the measurement pattern is a benefit that is explicitly stated and empirically verified. As highlighted in the introductory section and pointed out by the reviewer, we further point out that his benefit is highly relevant in a clinical MRI setting, where scan parameters may change from patient to patient. For example, changing the field of view (to accommodate different patient sizes), while keeping the same resolution and scan time will result in a different acceleration factor, thus a different measurement distribution. While an ensemble of end-to-end models could be trained, it is unclear at what granularity this should be done. Thus, we argue that a model that is robust to measurement shifts is valuable from a clinical use perspective.
>
> **Concern 4: "_...Most of other methods they compare against in Figure 4c also see robust to such image distribution mismatch. I wonder what is the significance of the proposed method in that case?_"**
>
>
> The central finding of our paper is that CSGM style generative models can be robust to changes in image and measurement distributions for compressed sensing MRI in a range of clinically relevant settings. Prior work [3] has shown robustness of other models for _particular_ distribution shifts, also confirmed by our Figure 4c. Additionally, we demonstrate that there exist shifts for which CSGM type models are more robust (see Figure 2 and the other sub-figures in Figure 4) compared to other proposed methods. In the next revision of our paper, we will clarify this point further to avoid any confusion.
>
> **Concern 5: "_The paper does not compare with the original CSGM framework in which you use a trained generative model and solve an optimization problem similar to the one in Eq. 2._"**
>
> We trained StyleGAN2 models on fastMRI data and found that the original approach is not competitive. We will emphasize this finding in the final version of the paper and include representative examples in the Appendix.
>
> **Concern 6: "_SSIM is a metric often used for natural image quality assessment. It is not necessarily a good metric for MR images. Please report PSNR as well._"**
>
> SSIM, while imperfect, is the main metric used in the fastMRI challenge [4]. Furthermore, we have already reported the PSNR values for all experiments in Figure 19, Appendix G.
>
> **Concern 7: "_Theoretical results: I feel there is a mismatch between the theoretical and empirical results in the paper. Gaussian measurements do not exist in MRI, so what is the purpose of discussing those results? Theorem 3.4 talks about general operators, but it is also a generic result that does not take into account any specific component of the proposed generative model-based method._"**
>
> These results show that posterior sampling is a principled algorithm that is provably near-optimal with _either_ Fourier measurements and no distribution shift _or_ Gaussian measurements and distribution shift. There is indeed a mismatch from the empirical results -- which feature both Fourier measurements and distribution shift -- but as noted in the remark below Theorem 3.4, it is unclear what theorem one could hope to prove in that setting.  Our theory does give strong motivation for
> the use of posterior sampling as an algorithm with non-Gaussian measurements and with distribution shift, and our empirical results show that posterior sampling succeeds where prior CSGM style algorithms have failed.
>
> Theorem 3.4 does take into account one specific component of the proposed method: that Langevin dynamics approximate posterior sampling.  Prior work on generative reconstruction has mostly been based on MAP/Maximum Likelihood estimates for recovery, for which the conclusion of Theorem 3.4 would be false.
>
>
> **Concern 8: Limitations and Societal Impact**
>
> We thank the reviewer for raising this important issue and would like to emphasize two types of limitations: algorithmic limitations and study limitations. We discuss algorithmic limitations of our method in terms of computational runtime in Section 4.1 (bottom of page 7).
>
> We recognize that our work also has study limitations, as our reconstructed images have not been reviewed by radiologists or clinicians and this is an important next step after a proof-of-principle feasibility study. We will emphasize this point in the revision of our paper.
>
> **References**
>
> [1] Kelkar, V.A., Anastasio, M.. (2021). Prior Image-Constrained Reconstruction using Style-Based Generative Models. Proceedings of the 38th International Conference on Machine Learning. Available from http://proceedings.mlr.press/v139/kelkar21a.html.
>
> [2] Dominik Narnhofer, Kerstin Hammernik, Florian Knoll, and Thomas Pock. Inverse gans for
> accelerated mri reconstruction. In Wavelets and Sparsity XVII. International Society for Optics and Photonics, 2019.
>
> [3] Darestani, M.Z., Chaudhari, A.S., Heckel, R.. (2021). Measuring Robustness in Deep Learning Based Compressive Sensing. _Proceedings of the 38th International Conference on Machine Learning_, in _Proceedings of Machine Learning Research_ 139:2433-2444. Available from http://proceedings.mlr.press/v139/darestani21a.html.
>
> [4] fastMRI challenge. https://fastmri.org/.

---

> > ### Comment · Reviewer_pPk8 · 2021-08-26
> > **follow up after author response**
> >
> > I appreciate the response authors provided to me and other reviewers. Some answers have helped me better understand the paper, but I still have some (major) concerns that I list below.
> >
> > - Concern 3 (shift in measurement distribution is irrelevant to this paper): Authors have emphasized that the measurement distribution shift can be an issue both in the paper and in the response. My concern is that distribution shift in measurements is irrelevant in the context of this paper. This paper is about compressed sensing, MRI, and generative models. I think anyone working in any of these areas would (or should) know that a model-based optimization problem like the one listed in Eq. 2 can use any combination of the sampling patterns and generative priors. The quality of reconstruction will of course depend on the combination of sampling, priors, and the specific signal under reconstruction. It is unclear to me if the posterior sampling  has anything to do with the robustness to sampling shifts, but that is the impression given by the abstract and the paper. Presenting measurement distribution shift in this manner seems unfair to me.
> >
> > - Concern 4 (robustness to shift in image distribution): Results in Figure 2 and Figure 4 show that the proposed method gives better quality than other methods in some cases, but in other cases quality is similar. I would like to see a discussion on what is the source of improvement? Is this the posterior sampling or the use of the score-based generative model or something else? A comparison with CSGM (see below) may help answer this question as well.
> >
> > - Concern 5 (comparison with CSGM): Your response is vague and I did not understand what is the problem. Is it that the trained network was not good even for reconstructing images similar to the ones in the training set or that the reconstruction was poor compared to the proposed approach. I think the comparison with CSGM is important.
> >
> > - Concern 7 (theoretical results): I am not fully convinced by your explanation, but I agree that Theorem 3.4 provides some connection with the MRI. Overall, I feel the theoretical analysis is disconnected from the MRI reconstruction problem, which is apparently the main focus of the paper. At this point, it is a minor issue for me.
> >
> > - The sampling regime is not compressive?: I did not raise this concern in my earlier review, but I saw this in another review. I think most of the results in the paper are not for compressive regime. It is important for authors to clarify the definition of the acceleration factor and the number of coils used for each data. I did not see this discussion in the main paper or the supplementary material. If you have 15 coils and you subsample the data from each of them by a factor of 4, then a SENSE-type method (which solves a least squares problem) can usually provide a good reconstruction. I think the authors should also include the results for the least-squares reconstruction (SENSE) for comparison.

---

> > > ### Author Response · Authors · 2021-08-27
> > > **Response to Reviewer pPk8's follow-up**
> > >
> > > Thank you very much for the follow-up comment. Given that some questions are inter-related, please see our categorized replies below.
> > >
> > > **Comparison to CSGM**
> > >
> > > The problem with existing CSGM approaches is that they perform poorly in the MRI setting. To date there is no reproducible and successful application of this framework for multi-coil MRI in-distribution, let alone with anatomical shifts. Our previous comment meant that we have tried and failed to get the standard CSGM approach to work satisfyingly.
> > >
> > > To give concrete numbers: in the setting of Figure 4a of our paper, in-distribution brain with R=4, our approach obtains an average PSNR of 37 dB (Appendix G, Fig. 17a), while CSGM gets an average of only 25.1 dB. The performance is even worse under anatomy shifts. For knee scans, all algorithms in the paper have PSNR values of at least 23 dB at R=4, whereas CSGM reconstructions have an average PSNR of 9.1 dB. This trend is similar at higher acceleration factors. We will include this comparison in the final version of the paper.
> > >
> > > **Robustness to measurement shifts**
> > >
> > > Our algorithm performs better under measurement shifts than all other MRI reconstruction methods considered in this work.
> > >
> > > Our experiments demonstrating this are an important contribution to our paper.
> > > Suppose we had not included the sampling pattern experiments, and we had simply claimed this superiority as an obvious implication of our approach. Can the reviewer really claim that other reviewers and area chair would all take this on faith, and not raise the lack of validating experiments as a major concern?
> > >
> > > **Posterior sampling vs score-based model**
> > >
> > > As we understand it, the question is: we use posterior sampling with a score-based model. What would happen if we did MAP with a score-based model, or posterior sampling with a different model like StyleGAN2? The exact behavior depends on how you implement these, but in our experience both cases would give noticeably worse results even in-distribution. Of course, we cannot rule out that there exists some setting of hyperparameters that performs well. This is why our theoretical justification of posterior sampling is useful: it shows that, unlike MAP, posterior sampling is competitive in-distribution for any distribution and any measurement pattern (including MRI).
> > >
> > > We chose posterior sampling for this theoretical justification. We chose the score-based model because we found it much easier to work with than StyleGAN2. Our algorithm has just four hyperparameters: one for the initial value of the step size, two for the initial \& final value of the annealing noise, and finally one to account for noise in the measurements. By contrast, for StyleGAN2 even inverting fully observed images is non-trivial; the original paper has an entire section devoted to the numerous heuristics involved.
> > >
> > > **Comparison to SENSE and compressive regime**
> > >
> > > Below we give a table of the PSNR performance of the Langevin approach and SENSE on brain and knee scans, at each acceleration factor, in the setting of Figure 4. At all acceleration factors, SENSE is inferior to our approach, especially for the in-distribution setting (brain).
> > >
> > > |      Acceleration    | 2 | 4 | 8 | 12 |
> > > |-----|------|------|-----|-----|
> > > | Knee-SENSE  | 29.7 | 27.6 | 24.8 | 24.4 |
> > > | Knee-Ours   | 33.1 | 30.1 | 26.8 | 24.7 |
> > > | Brain-SENSE | 32.3 | 28.7 | 23.4 | 22.6 |
> > > | Brain-Ours  | 40.0 | 36.8 | 32.1 | 30.8 |
> > >
> > > This is not surprising, as it is very well-documented that SENSE and other parallel imaging methods fail in 2D MRI scanning at acceleration factors R=4 and higher, even with 15 coils. See Figures 3-7 and Table 1 ("CG-SENSE") in the seminal paper [1] for example, which uses the same scanning protocol as the fastMRI knee dataset as we use in our work.
> > >
> > > Regarding the compressive regime, note that for 2D MRI the acceleration can only be applied in the phase encode direction, and hence the redundancy provided by the coils is low due to the physical layout of the coils. This is again a well-documented phenomenon in the MRI literature [2, 3]. We will include this discussion in the paper.
> > >
> > > **References**
> > >
> > > [1] Hammernik, K., Klatzer, T., Kobler, E., Recht, M.P., Sodickson, D.K., Pock, T. and Knoll, F. (2018), Learning a variational network for reconstruction of accelerated MRI data. Magn. Reson. Med., 79: 3055-3071. https://doi.org/10.1002/mrm.26977.
> > >
> > > [2] Weiger M, Pruessmann KP, Boesiger P. 2D SENSE for faster 3D MRI. MAGMA. 2002 Mar;14(1):10-9. doi: 10.1007/BF02668182. PMID: 11796248.
> > >
> > > [3] Breuer, F.A., Blaimer, M., Mueller, M.F., Seiberlich, N., Heidemann, R.M., Griswold, M.A. and Jakob, P.M. (2006), Controlled aliasing in volumetric parallel imaging (2D CAIPIRINHA). Magn. Reson. Med., 55: 549-556. https://doi.org/10.1002/mrm.20787.

---

> ### Author Response · Authors · 2021-08-20
> **Follow-up to author response**
>
> Thank you again for your review. As a gentle follow-up, based on your reading of our initial author response and the other reviews, do you have any additional concerns or comments we can address?
>
> Based on the reviewer comments and the response from Ethics Reviewer Kb9Y and Reviewer pPk8, we are conducting a preliminary image quality assessment with two board-certified radiologists and one faculty member who uses neuroimaging for their research. The three external experts are not involved with our study and will perform the image quality assessment blindly (i.e. without knowing which reconstruction algorithm was used for each image). We will post the results in the comments here, when it is complete. We emphasize that this a very preliminary experiment and not a replacement for a future detailed study that evaluates clinical diagnostic quality and this will be clear in the paper.

---

> ### Author Response · Authors · 2021-08-26
> **Follow-up to Reviewer pPk8 regarding Expert Quality Assessment**
>
> We thank Reviewer pPk8 once again for their valuable feedback and would like to kindly point out that we have submitted the results of a preliminary image quality assessment experiment as a top-level comment.
>
> We have conducted this assessment with two board-certified radiologists and a faculty member that are not involved in our research, and have found that our algorithm is ranked as best for knee scans, and ties with the baselines for abdominal and brain scans, supporting our robustness claims in the paper. For more details, please see our high-level comment and we welcome any feedback.

---

### Review · Ethics_Reviewer_Kb9Y · 2021-08-09

**Recommendation:**

I recommend that:
A) if the area chair is familiar with prior work showing that SSIM has external real-world validity in measuring phenomena of concerns to radiologists, then the the area chair should request that the authors cite this work
Otherwise:
B) I recommend that the area chair request that the authors acknowledge that i) a limitation of their approach to be lack of validation with real-world radiologists, and that ii) explicitly state that the techniques should not be used in real world medical scenarios until such validation is done. I also recommend that the chair urge the authors to conduct such a validation study with radiologists and publish its results regardless of whether they are positive or negative.

**Ethical Issues:**

Yes

**Ethics Review:**

This paper was flagged for ethics review only by pPk8, raising the following issues:
1) "MRI is a popular imaging scheme and we need thorough analysis and test to make claims about generalization/accuracy of the proposed methods."
2) "I did not find an adequate discussion on ethical concerns in the paper."
3) "The paper actually claim that they are the first to show results on clinical data, but they probably should rephrase it as publicly available real data."
Regarding (1), I agree that medical data raises heightened concerns about consequences of misuse. I am not an expert in this area, so I cannot judge whether the authors' analysis in insufficiently thorough.

To state my concern succinctly: I would not want these techniques to be used in real world medical applications until further validation studies are complete.

To elaborate: my main concern is that radiology is a highly technical and consequential domain of medical specialization. In particular my concerns about external validity of metrics. It is not obviously clear to me that the SSIM metric advocated by [86] accurately captures the elements of images that radiologists find most informative. I am not sure if this has been established by prior work, as this is not my area of expertise. If this has been established by prior work, then such work should be cited. Otherwise, what seems to be missing is a validation study where the images are shown to domain experts (i.e. radiologists). In the absence of such a study, the paper should be explicit about its limiting assumption that SSIM accurately captures the real-world needs of radiologists.

---

> ### Author Response · Authors · 2021-08-20
> **Response to Ethics Reviewer Kb9Y**
>
> We thank the reviewer for their thoughtful feedback regarding ethical concerns.
>
> **Concern 1: "To state my concern succinctly: I would not want these techniques to be used in real world medical applications until further validation studies are complete."**
>
> We absolutely agree. Our paper is a proof-of-principle technical study on the method. Further validation studies involving radiologists are absolutely necessary before making steps towards clinical use.
> We will add a detailed statement to the paper that the techniques should not be used until clinical validation studies have been performed.
>
> **Concern 2: "It is not obviously clear to me that the SSIM metric advocated by [86] accurately captures the elements of images that radiologists find most informative"**
>
> It is true that SSIM and PSNR are not sufficient to address diagnostic quality and radiologist perspective; however, SSIM and PSNR are currently the established metrics used in the reconstruction community prior to the important next step of validation studies. We note that, although SSIM alone is not sufficient, it is highly correlated to radiologist score, as explained in [1], which conducted a study comparing SSIM to radiologist score:
>
> "A case-wise breakdown of the ranks for all 3 finalists and all rated cases is shown in Figure 5. For second and third- place metrics as rated by SSIM, radiologist assessment was discordant between the two methods. However, in 16 out of 18 cases the highest SSIM score within the finalists’ batches also received the highest radiologists’ rating. A similar relation- not shown here - was found for the other used metrics such as normalized mean-squared error (NMSE) and peak signal- to-noise ratio (PSNR)."
>
> "In terms of radiologist evaluations, despite the drawbacks of SSIM and RSS ground truths, we observed a correlation between radiologist scores and SSIM scores for large SSIM separations."
>
> [1] M. J. Muckley et al., "Results of the 2020 fastMRI Challenge for Machine Learning MR Image Reconstruction," in IEEE Transactions on Medical Imaging, doi: 10.1109/TMI.2021.3075856.
>
> In any case, we will include a clarification of the known limitations of the simple metrics we use in our paper in addition to the disclaimer that this only the first paper introducing a novel method.
>
> **Concern 3: "What seems to be missing is a validation study where the images are shown to domain experts (i.e. radiologists). In the absence of such a study, the paper should be explicit about its limiting assumption that SSIM accurately captures the real-world needs of radiologists."**
>
> Thank you for bringing this up. We will add explicit discussion about the limitations of SSIM and explicitly state that the techniques should not be used in real world medical scenarios until such a validation is done.
>
> Based on the reviewer comments and the response from Ethics Reviewer Kb9Y and Reviewer pPk8, we are conducting a preliminary image quality assessment with two board-certified radiologists and one faculty member who uses neuroimaging for their research. The three external experts are not involved with our study and will perform the image quality assessment blindly (i.e. without knowing which reconstruction algorithm was used for each image). We will post the results in the comments here, when it is complete. We emphasize that this a very preliminary experiment and not a replacement for a future detailed study that evaluates clinical diagnostic quality and this will be clear in the paper.

---

> ### Author Response · Authors · 2021-08-26
> **Follow-up to Ethics Reviewer Kb9Y**
>
> We thank Ethics Reviewer Kb9Y once again for their valuable suggestions and would like to kindly point out that we have submitted the results of a preliminary image quality assessment experiment as a top-level comment.
>
> We have conducted this assessment with two board-certified radiologists and a faculty member that are not involved in our research, and have found that our algorithm is ranked as best for knee scans, and ties with the baselines for abdominal and brain scans, supporting our robustness claims in the paper. For more details, please see our high-level comment and we welcome any feedback.

---

### Review · Ethics_Reviewer_7dWs · 2021-08-11

**Recommendation:**

I recommend including a discussion of ethical concerns, as detailed above in the Ethics Review.

**Ethical Issues:**

Yes

**Ethics Review:**

The paper is missing a discussion of ethical concerns. Since the application scenario considered in this paper may impact human lives, it is important to consider and comment on potential societal impacts.

Firstly, it would be worth commenting on potential issues related to discrimination, referencing prior work on algorithmic fairness. Specifically, it is possible that the quality of the reconstructed images varies across protected attributes, such as gender or race. An example of algorithmic fairness issues in medical imaging can be found in recent work by Larrazabal et al. [1]. Additionally, it would be useful to emphasize that the quality of the reconstructed images has not been evaluated by medical professionals. I suggest discussing these limitations.

[1] Larrazabal, A. J., Nieto, N., Peterson, V., Milone, D. H., & Ferrante, E. (2020). Gender imbalance in medical imaging datasets produces biased classifiers for computer-aided diagnosis. Proceedings of the National Academy of Sciences

---

> ### Author Response · Authors · 2021-08-20
> **Response to Ethics Reviewer 7dWs**
>
> We thank the reviewer for their thoughtful feedback regarding ethical concerns.
>
> **Concern 1: "It would be worth commenting on potential issues related to discrimination, referencing prior work on algorithmic fairness. Specifically, it is possible that the quality of the reconstructed images varies across protected attributes, such as gender or race."**
>
> Indeed algorithmic fairness across protected attributes is important to consider. As the FastMRI and Stanford MRI datasets are fully anonymized, and the data do not contain information regarding protected attributes such as gender or race, we cannot claim that the data are or are not well-balanced with respect to these considerations. Note that this valid concern applies not only to our work, but also to the many other works that use these publicly available datasets. We will revise our paper to include a statement regarding potential issues related to discrimination and reference the work of [1]
>
> [1] Larrazabal, A. J., Nieto, N., Peterson, V., Milone, D. H., and Ferrante, E. (2020). Gender imbalance in medical imaging datasets produces biased classifiers for computer-aided diagnosis. Proceedings of the National Academy of Sciences
>
> **Concern 2: "It would be useful to emphasize that the quality of the reconstructed images has not been evaluated by medical professionals. I suggest discussing these limitations."**
>
> We thank the reviewer for pointing this out and we will emphasize that the quality of the reconstructed images has not been evaluated by medical professionals.
>
> Based on the reviewer comments and the response from Ethics Reviewer Kb9Y and Reviewer pPk8, we are conducting a preliminary image quality assessment with two board-certified radiologists and one faculty member who uses neuroimaging for their research. The three external experts are not involved with our study and will perform the image quality assessment blindly (i.e. without knowing which reconstruction algorithm was used for each image). We will post the results in the comments here, when it is complete. We emphasize that this a very preliminary experiment and not a replacement for a future detailed study that evaluates clinical diagnostic quality and this will be clear in the paper.

---

### Author Response · Authors · 2021-08-10
**General Response to All Reviewers**

We thank the reviewers for their feedback and careful reading of the paper. We are delighted that the reviewers found our work to be a significant development[Reviewers zR76, nt1J], original[Reviewers  zR76, nt1J], well-described and well-evaluated[Reviewers zR76, nt1J, HfL5], and our theoretical results interesting[Reviewers zR76, nt1J, HfL5]. We also appreciate the constructive criticism highlighted by the reviewers regarding our
theoretical results[Reviewers pPk8, zR76],
exposition of reported metrics [Reviewers zR76, nt1J],
significance of certain experiments[Reviewers pPk8, nt1J],
and societal \& ethical impact[Reviewers pPk8, nt1J].

We have addressed individual reviewer concerns as replies to the reviews.


**@Reviewer zR76, Reviewer nt1J: Concerns regarding reported SSIM metrics due to choice of RSS vs. MVUE**

The reviewers raised concerns regarding the differences in ranking between our submission and the literature. This is due to the differences in MVUE and RSS images, which has been discussed in prior work [KH]. For a more detailed response, please see our response to Reviewer zR76, Concern 1 & 2, and our response to Reviewer nt1J, Concern 5.

We will include a more detailed discussion in the main paper and supplementary material.

[KH] Hammernik, K, Schlemper, J, Qin, C, et al. Systematic evaluation of iterative deep neural networksfor fast parallel MRI reconstruction with sensitivity-weighted coil combination. Magn Reson Med. 2021; 86:1859– 1872. https://doi.org/10.1002/mrm.28827

**@Reviwer pPk8, Reviewer nt1J: Concerns regarding significance of robustness to k-space sampling patterns**

The reviewers claimed that the parameters in the measurement process do not change in the clinic, and hence questioned the significance of the mask shift in our experiments. As detailed in our response to Reviewer pPk8, Concern 3 & 4, and Reviewer nt1J, Concern 3, there are cases where the acquisition parameters change, and this changes the k-space sampling pattern. We will discuss this more carefully in the main paper.

**@Reviewer pPk8, Reviewer nt1J: Ethical concerns regarding evaluation by radiologists**

In the original submission, we had focused on the algorithmic limitations of our work. We thank the reviewers for pointing out the ethical concerns, and we agree that diagnostic evaluation by a clinician is an important next step following this proof-of-principle feasibility study. We will emphasize this point in the revision of our paper.

**@Reviewer pPk8, Reviewer zR76: Caveats and concerns regarding theoretical results.**

We thank the reviewers for their constructive criticism. Our theoretical results show that posterior sampling is a principled algorithm that is provably near-optimal with *either* Fourier measurements and no distribution shift *or* Gaussian measurements and distribution shift. In our original submission, we discussed some of the caveats of our results in the remark under Theorem 3.4. We will also include the caveats pointed out by the reviewers. For a more detailed response, please see our response to Reviewer pPk8, Concern 7, and Reviewer zR76, Concern 5, Part 1 & 2.

---

### Author Response · Authors · 2021-08-26
**General Response Regarding Expert Quality Assessment**

We would like to once again thank all reviewers and ethics reviewers for their valuable suggestions. As we have stated in our previous response, based on responses from Reviewer pPk8 and Ethics Reviewer Kb9Y, we have performed a preliminary image quality assessment experiment with two board-certified radiologists and a faculty member that uses neuroimaging in their research.

The three external experts were not involved with our research and have performed the image quality assessment blindly. Each of them was presented with ten scans from the following anatomies and scan parameters: abdominal scans, knee scans and brain scans with a horizontal readout direction, leading to a total of 30 quality assessment questions. Note that all anatomies represent test-time distributional shifts in at least one aspect.

In each question, the experts were shown four images:
* The fully-sampled reference image, explicitly marked as "Reference".
* The results of three reconstruction algorithms at acceleration factor R=4: MoDL, ConvDecoder and our method. The order of the reconstructions was shuffled for each question, and the reconstructions were labeled as "1", "2" and "3".

We chose to compare with MoDL since this method had the best overall quantitative and qualitative (according to our own pre-assessment) performance, and with ConvDecoder, since this method was reported to be robust in prior work [1]. The participants were instructed to rank the three reconstructions from best to worst quality, while using the "Reference" image as a perceptual guideline. The table below shows the average and standard deviation (in parentheses) of the ranking for each anatomy, obtained using a total of 30 data points (3 participants x 10 scans per anatomy):

|                               |  MoDL   |          ConvDec          | Ours |
|-------------------------------|---------|------------------------|----------------|
| Knee                          | 1.87 (0.34) | 2.97 (0.18) | 1.17 (0.45) |
| Abdomen                       | 1.87 (0.76) | 2.17 (0.93) | 1.97 (0.71) |
| Brain                         | 2.00 (0.82) | 2.07 (0.77) | 1.93 (0.85) |

In the table above, a lower ranking is better, the best possible ranking is 1, and the worst 3. We draw the following conclusions:
* Participants consistently ranked our method as best on the knee scans, which supports the distributional shift robustness claimed in the main paper, and detailed in Appendices C and G.
* Participants did not perceive a significant difference between all methods when applied to abdominal or brain scans with a horizontal phase encode direction. In the brain case, this supports the qualitative results shown in Appendix B.2, Figure 8. In the abdominal case, this partially correlates with Figure 4c, regarding the quantitative tie between our approach and MoDL.

To quantify the statistical significance of the above results, we perform a Wilcoxson Rank Sum test to determine if the rankings of different algorithms are drawn from different populations. We evaluate if our proposed method leads to different rankings than MoDL and the ConvDecoder, and show the p-values in the table below:

|                               |  Ours vs. MoDL   | Ours vs. ConvDec |
|-------------------------------|---------|------------------------|
| Knee                          | 1.53e-10 | 2.77e-6 |
| Abdomen                       | 0.610  | 0.340 |
| Brain                         | 0.767 | 0.550 |

The results show a significant difference in the case of knees, while no significant difference is present for abdomen and brain. Finally, to evaluate inter-observer agreement between the three reviewers, we calculated the intraclass correlation (ICC) coefficient separately for each anatomy by aggregating the ten questions related to that anatomy and evaluating the ICC2 coefficient [2] in a pairwise manner at a 5\% significance level.

The results are shown in the table below, where we also include the p-value and the 95\% confidence interval for the ICC2 estimate. This indicates that there exists a very strong consensus regarding the ranking on the knee anatomy, while for abdomen and brain this consensus is much weaker, which together with the previous table indicates that the images were considered equivalent:

|                               |  ICC2   |          p-value          | 95% CI |
|-------------------------------|---------|------------------------|----------------|
| Knee                          | 0.980 | 0.0004 | [0.81 1.  ] |
| Abdomen                       | -0.222 | 0.576 | [-0.89  0.92] |
| Brain                         | -0.818 | 0.907 | [-0.98  0.59] |

The preliminary image quality assessment gives additional evidence (in addition to the quantitative metrics of SSIM and PSNR) that our method maintains robustness to distribution shifts at test time. As our quantitative results show, other methods maintain robustness in some but not all cases. Due to time limitations, we were not able to ask the reviewers to evaluate every algorithm and every distribution shift including different levels of acceleration. We stress that this preliminary study is not a substitute for a rigorous clinical evaluation which is necessary before considering using our proposed method in a clinical setting.

We again want to thank all the reviewers for their valuable feedback as we believe these results contribute to strengthening our claims.

**References**

[1] Darestani, Mohammad Zalbagi, Akshay Chaudhari, and Reinhard Heckel. "Measuring Robustness in Deep Learning Based Compressive Sensing", ICML 2021.

[2] https://pingouin-stats.org/generated/pingouin.intraclass_corr.html.

---

### Decision · Program_Chairs · 2021-09-27

**Decision:**

Accept (Poster)

**Comment:**

The authors provide the first demonstration that Compressed Sensing with Generative Priors can be a competitive approach for MRI reconstruction relative to end-to-end, L1, and untrained neural network methods.  The authors additionally demonstrate that the method is more robust than baselines in the context of certain distribution shifts between training and inversion.  The paper also provides novel theoretical guarantees about distributional robustness of posterior sampling approach.

There was some concern from one of the reviewers about consistency with the literature about the performance of baseline methods, likely due to training being on RSS images with fine-tuning on MVUE images.  The authors should provide commentary about this issue, as discussed with the reviewers during the rebuttal.

The authors should add additional comments that the paper should not be used for medical purposes without subsequent study by medical professionals, as per the ethics review.